# Diversity Is All You Need for Contrastive Learning: Spectral Bounds on Gradient Magnitudes

**Peter Ochieng**[*]
Department of Computer Science
University of Cambridge
po304@cam.ac.uk

## Abstract

We derive non-asymptotic spectral bands that bound the squared InfoNCE gradient norm via alignment, temperature, and batch spectrum, recovering the $1/\tau^2$ law and closely tracking batch-mean gradients on synthetic data and ImageNet. Using effective rank $R_{\text{eff}}$ as an anisotropy proxy, we design spectrum-aware batch selection, including a fast greedy builder. On ImageNet-100, Greedy-64 cuts time-to-67.5% top-1 by 15% vs. random (24% vs. Pool–P3) at equal accuracy; CIFAR-10 shows similar gains. In-batch whitening promotes isotropy and reduces 50-step gradient variance by $1.37\times$, matching our theoretical upper bound.

## 1 Introduction

Contrastive learning is a *label-efficient* paradigm for representation learning: it pulls together *positive* views of the same instance while pushing apart *negatives* in the mini-batch, uncovering latent structure without class labels. Yet this tug-of-war is delicate: weak positives stall learning, whereas overly similar negatives drive gradients toward zero, even in frameworks that rarely collapse in practice such as SimCLR (Chen et al., 2020a) and MoCo (He et al., 2020b). Most prior work tackles this at the *pair level* (e.g., hard-negative mining and debiased losses), but overlooks a batch-level signal: the *spectrum* of the embedding cloud. If that cloud is too *narrow* (high anisotropy; low effective rank), negatives become redundant; if too *wide* (near-isotropic with uniformly small pairwise similarities), the softmax distribution flattens. Empirically, we find training is fastest inside a *moderate-diversity window*—rich enough to avoid collapse, yet narrow enough to preserve directional signal.

We formalise and exploit this observation via three contributions:

1. **Gradient-norm spectral band.** We derive sharp, non-asymptotic bounds on the *squared* InfoNCE gradient norm, decomposing contributions from positive alignment, finite-sample variance, and batch anisotropy. The band depends only on in-batch (or queue) second moments, making the analysis—and the resulting diagnostics—agnostic to whether negatives are drawn in-batch (SimCLR) or from a queue (MoCo).

2. **Diversity-aware batch construction.** We propose two lightweight samplers that keep training inside the diversity window: (i) a *pool* selector that targets high effective rank $R_{\text{eff}}$; and (ii) a streaming *Greedy-m* builder that adds the sample with the largest spectral-diversity gain. In our measurements, Greedy-$m$ adds $\lesssim 1\%$ on-GPU overhead, while large host-side pools can incur $\sim 8\,\text{ms}$/iteration on a V100 (see §B.3).

3. **Experiments from 100 to 1000 classes.** On ImageNet-100 our samplers reduce wall-clock time to $67.5\%$ top-1 by $\sim 15\%$ with no accuracy loss. On ImageNet-1k, linear-eval accuracy increases with $R_{\text{eff}}$ and plateaus once $R_{\text{eff}}/C \gtrsim 0.9$, consistent with the RankMe view that diversity gains saturate near this threshold (§B).

---

[*]Accepted as a Poster at the 39th Conference on Neural Information Processing Systems (NeurIPS 2025)

## 1.1 Notation

Let batch size $n > 2$ and embedding dimension $d$. Each embedding $z_i \in \mathbb{S}^{d-1} \subset \mathbb{R}^d$ has a designated positive $z_{i^+}$. Write the set of *others* as $\mathcal{S}_i := \{1, \ldots, n\} \setminus \{i\}$, $|\mathcal{S}_i| = n-1$, and the *negatives* as $\mathcal{N}_i^- := \mathcal{S}_i \setminus \{i^+\}$, $|\mathcal{N}_i^-| = n-2$. Let $Z = [\, z_1; \ldots; z_n \,] \in \mathbb{R}^{n \times d}$ collect rows $z_i^\top$, and define pairwise similarities $s_{ij} = z_i^\top z_j \in [-1, 1]$ with softmax temperature $\tau > 0$.

**Second moments.** Define the batch second moment and the per-anchor "others"/"negatives-only" moments: $\hat{\Sigma} := \frac{1}{n} \sum_{j=1}^n z_j z_j^\top$, $\tilde{\Sigma}_i := \frac{1}{n-1} \sum_{j \in \mathcal{S}_i} z_j z_j^\top$, $\tilde{\Sigma}_i^- := \frac{1}{n-2} \sum_{j \in \mathcal{N}_i^-} z_j z_j^\top$. Under unit norms, $\text{tr}(\hat{\Sigma}) = \text{tr}(\tilde{\Sigma}_i) = \text{tr}(\tilde{\Sigma}_i^-) = 1$. Exact relations: $\tilde{\Sigma}_i = \frac{n}{n-1} \left( \hat{\Sigma} - \frac{1}{n} z_i z_i^\top \right)$, $\tilde{\Sigma}_i^- = \frac{n}{n-2} \left( \hat{\Sigma} - \frac{1}{n}(z_i z_i^\top + z_{i^+} z_{i^+}^\top) \right)$. Hence the spectral bounds used later:

$$\lambda_{\max}(\tilde{\Sigma}_i) \leq \frac{n}{n-1} \lambda_{\max}(\hat{\Sigma}), \quad \lambda_{\max}(\tilde{\Sigma}_i^-) \leq \frac{n}{n-2} \lambda_{\max}(\hat{\Sigma}).$$

We write $\hat{\sigma} := \lambda_{\max}(\hat{\Sigma})$, $\sigma_i := \lambda_{\max}(\tilde{\Sigma}_i)$, $\sigma_*^{(i)} := \lambda_{\max}(\tilde{\Sigma}_i^-) \in [1/d, 1]$ (minimum at isotropy).

## 1.2 Bounding $\|\nabla_{z_i} \mathcal{L}_i\|^2$ — Spectral Gradient Band

We show that the per-sample InfoNCE gradient is confined to a *spectral band* whose width is governed by alignment, finite-sample noise, batch anisotropy, and temperature.

**Setup & assumptions.** Let $N^- = n-2$, $\tilde{\Sigma}_i^- := \frac{1}{N^-} \sum_{j \in \mathcal{N}_i^-} z_j z_j^\top$, and $\sigma_*^{(i)} := \lambda_{\max}(\tilde{\Sigma}_i^-)$. With unit norms and the Löwner order, $\tilde{\Sigma}_i^- = \frac{1}{N^-}(n\hat{\Sigma} - z_i z_i^\top - z_{i^+} z_{i^+}^\top) \preceq \frac{n}{n-2} \hat{\Sigma} \Rightarrow \sigma_*^{(i)} \leq \frac{n}{n-2} \hat{\sigma}$, where $\hat{\Sigma} = \frac{1}{n} \sum_i z_i z_i^\top$ and $\hat{\sigma} = \lambda_{\max}(\hat{\Sigma})$. InfoNCE: $\mathcal{L}_i = -\log p_{ii^+}$ with $p_{ij} \propto e^{s_{ij}/\tau}$, $s_{ij} = z_i^\top z_j$, $\epsilon_i := 1 - p_{ii^+}$, and $M_i = \sum_k p_{ik} z_k$ so $\nabla_{z_i} \mathcal{L}_i = \tau^{-1}(M_i - z_{i^+})$. Expectations are over the mini-batch/augmentations. Assumptions: (A1) unit norms; (A2) negatives i.i.d. within the batch and independent of $z_i$ (no assumption on the positive pair).

**Theorem 1.1** (Gradient–Norm Spectral Band). *Under (A1)–(A2), for a softmax-smoothness constant $c > 0$,*

$$\mathbb{E}\big[\|\nabla_{z_i} \mathcal{L}_i\|^2\big] \leq \frac{3}{\tau^2} \Big( \mathbb{E}[\epsilon_i^2] + \frac{\mathbb{E}[(1-p_{ii^+})^2]}{N^-} \Big) + \frac{3\,\mathbb{E}[(1-p_{ii^+})^2]\,\sigma_*}{\tau^4} + \frac{3c\,\mathbb{E}[(1-p_{ii^+})^2]\,\sigma_*^2}{\tau^6}, \quad (1)$$

$$\mathbb{E}\big[\|\nabla_{z_i} \mathcal{L}_i\|^2\big] \geq \frac{(1-\rho)^2}{\tau^2}, \qquad \rho := \mathbb{E}\big[\langle M_i, z_{i^+}\rangle\big], \quad (2)$$

*where $\sigma_* := \mathbb{E}[\sigma_*^{(i)}]$ (or use the per-anchor form).*

**Sketch.** Decompose $\delta_i := M_i - z_{i^+} = A_i + B_i + C_i$ with $A_i = -\epsilon_i z_{i^+}$, $B_i = (1 - p_{ii^+})\bar{z}_i^-$, $\bar{z}_i^- := \frac{1}{N^-} \sum_{j \in \mathcal{N}_i^-} z_j$, and $C_i = (1 - p_{ii^+}) \sum_j (q_{ij} - \frac{1}{N^-}) z_j$ where $q_{ij} = p_{ij}/(1 - p_{ii^+})$. Then $\|\delta_i\|^2 \leq 3(\|A_i\|^2 + \|B_i\|^2 + \|C_i\|^2)$, $\mathbb{E}\|A_i\|^2 = \mathbb{E}[\epsilon_i^2]$, $\mathbb{E}\|B_i\|^2 = \mathbb{E}[(1-p_{ii^+})^2]/N^-$ by (A2), and a first-order Taylor of the negatives-only softmax around uniform logits (App. A.3) gives $\mathbb{E}\|C_i^{(1)}\|^2 \leq \tau^{-2} \mathbb{E}[(1-p_{ii^+})^2]\sigma_*$ and $\mathbb{E}\|C_i^{(2)}\|^2 \leq c\tau^{-4} \mathbb{E}[(1-p_{ii^+})^2]\sigma_*^2$. For the lower band, $\|M_i\| \leq 1$ implies $\|M_i - z_{i^+}\|^2 \geq (1 - \langle M_i, z_{i^+}\rangle)^2$; apply Jensen. Full details in App. A.1.

**Variants.** (i) *Bounded spectrum:* if $\sigma_*^{(i)} \leq \sigma_{\max}$, replace $\sigma_*$ by $\sigma_{\max}$. Using $\sigma_*^{(i)} \leq \frac{n}{n-2} \hat{\sigma}$ gives a batch-measurable ceiling. (ii) *High probability:* if $z_j$ are i.i.d. sub-Gaussian with $\text{tr}\,\Sigma = 1$, matrix concentration (App. A.2) yields $\lambda_{\max}(\hat{\Sigma}) \lesssim \frac{1}{d} + O(\sqrt{\frac{\log d}{n}})$ and hence $\sigma_*^{(i)} \leq \frac{n}{n-2} \lambda_{\max}(\hat{\Sigma})$; substitute into (1).

**Interpretation.** Softmax error $\epsilon_i$ and the sampling term $1/N^-$ set a baseline that decays with batch size. Anisotropy enters via $\sigma_*$ at orders $\tau^{-4}$ and $\tau^{-6}$; pushing toward isotropy (e.g., larger $R_{\text{eff}}$, whitening) tightens the ceiling. Higher alignment $\rho$ shrinks the floor $(1-\rho)^2/\tau^2$ without implying collapse.

## 1.3 Spectral Batch Selection

The upper band in Thm. 1.1 depends on the per–anchor negatives-only spectrum $\sigma_*^{(i)} = \lambda_{\max}(\tilde{\Sigma}_i^-)$ with $\tilde{\Sigma}_i^- = \frac{1}{N^-} \sum_{j \in \mathcal{N}_i^-} z_j z_j^\top$, $N^- = n - 2$. Computing $\sigma_*^{(i)}$ for every anchor is costly, so we use the batch second moment $\hat{\Sigma} = \frac{1}{n} \sum_{i=1}^n z_i z_i^\top$ (trace one under $\|z_i\|=1$) and its top eigenvalue $\hat{\sigma} = \lambda_{\max}(\hat{\Sigma})$ as a proxy. By Löwner order,

$$\tilde{\Sigma}_i^- = \frac{1}{N^-}\big(n\hat{\Sigma} - z_i z_i^\top - z_{i+} z_{i+}^\top\big) \preceq \frac{n}{n-2} \hat{\Sigma} \;\Rightarrow\; \sigma_*^{(i)} \le \frac{n}{n-2} \hat{\sigma},$$

so lowering $\hat{\sigma}$ tightens a uniform ceiling on all $\sigma_*^{(i)}$.

**Effective rank.** Let $R_{\text{eff}} := 1/\operatorname{tr}(\hat{\Sigma}^2) = \big(\sum_k \lambda_k^2\big)^{-1}$ (eigs $\lambda_k$ of $\hat{\Sigma}$). Since $\sum_k \lambda_k = 1$ and $\sum_k \lambda_k^2 \le \hat{\sigma}$, we get $\hat{\sigma} \ge 1/R_{\text{eff}}$: higher $R_{\text{eff}}$ (more isotropy) $\Rightarrow$ smaller $\hat{\sigma}$ and a tighter band.

**Policies.** Given a candidate pool $\mathcal{P}_t = \{Z^{(m)}\}_{m=1}^k$ with ranks $R^{(m)}$: *P1 (stability)* picks $\arg\max_m R^{(m)}$ to minimize $\hat{\sigma}$; *P2 (update-magnitude)* picks $\arg\min_m R^{(m)}$ to allow larger steps (higher variance risk); *P3 (balanced)* picks $\arg\min_m |R^{(m)} - R_\star|$ with $R_\star$ set by running 10th/90th percentiles (App. A.5). In practice: use P1 early (collapse risk), switch to P3 once $R_{\text{eff}}$ stabilizes, and deploy P2 sparingly to escape flats.

## 1.4 Greedy Element–Wise Spectral Builder

We assemble a batch incrementally to *decrease* the trace of the squared second moment, thereby *increasing* $R_{\text{eff}} = 1/\operatorname{tr}(\Sigma^2)$. Assume unit–norm rows.

**Objective.** For a partial batch $B$ of size $b$, let

$$\Sigma_B = \frac{1}{b} \sum_{z' \in B} z' z'^\top, \qquad t_B = \operatorname{tr}(\Sigma_B^2), \qquad q_B(z) = z^\top \Sigma_B z = \frac{1}{b} \sum_{z' \in B} \langle z, z' \rangle^2.$$

Lemma A.3 (App. A.6) gives the one–step update for a unit–norm candidate $z$:

$$\operatorname{tr}(\Sigma_{B\cup\{z\}}^2) = \frac{b^2 t_B + 2b\, q_B(z) + 1}{(b+1)^2}, \quad \Rightarrow \quad \Delta t := \operatorname{tr}(\Sigma_{B\cup\{z\}}^2) - t_B = \frac{2b\,(q_B(z) - t_B) + (1 - t_B)}{(b+1)^2}.$$

Thus smaller $q_B(z)$ yields smaller (more negative) $\Delta t$.

**Greedy rule (Greedy–$m$).** At each step select

$$z^\star = \arg\min_{z \in \mathcal{C}} q_B(z),$$

where $\mathcal{C}$ is a probe set of size $m$ (from a pool or stream). For unit–norm rows this is equivalent to maximizing the Frobenius diversity $\Delta_F(z \mid B) = \|\Sigma_B - zz^\top\|_F^2 = t_B - 2q_B(z) + 1$.

**Cost.** With cached dot products against $B$, evaluating $q_B(z)$ costs $O(b)$ per candidate; a probe of size $m$ costs $O(mb)$ per step (or $O(mbd)$ without caching). We update $t_B$ via Lemma A.3 in $O(1)$ after each selection. A practical Greedy–$m$ realisation (initial seeds, Gram maintenance, streaming) is given in Algorithm 2 (App. A.6).

# 2 Experiments

We evaluate on standard self-supervised benchmarks with SimCLR Chen et al. (2020a) and MoCo v2 He et al. (2020a). Unless noted, we follow the original hyperparameters/architectures/augmentations; the *only* change is our spectrum-aware batch selection (§1.3–1.4). **Datasets:** ImageNet-100 (main), CIFAR-10 / Oxford Pets (linear probe). **Backbones:** ResNet-18 (default), ResNet-50 (variant). **Batch/Temp:** $n=256$, $\tau=0.2$ (SimCLR) unless stated. **Metrics:** final top-1 and *epochs to threshold* (ImageNet-100: 67.5%/70%). Significance via paired $t$-tests (95% CIs). Our main results use SimCLR; MoCo-v2 ablations and queue robustness are in App. A.11. Policy details (P1–P3), the Greedy–$m$ builder, pool sizing, and overheads are in App. A.5–A.6; spectrum metrics and evaluation protocols (e.g., RankMe, $R_{\text{eff}}$) are in App. B.

## 2.1 Synthetic Gradient–Band Verification

We validate the analytical upper/lower bounds on $\|\nabla_{z_i}\mathcal{L}_i\|^2$ (§1.2, Thm. 1.1) on controlled synthetic data, using a *per-batch plug-in* variant of the band (we replace expectations by measured batch quantities). Draw $x_i \sim \mathcal{N}(0, I_d)$ and set $z_i := \frac{Ax_i}{\|Ax_i\|}$, $A := U\Lambda^{1/2}$, $\Lambda := \mathrm{diag}(\lambda_1, \ldots, \lambda_d)$, $\sum_j \lambda_j = 1$, with $U$ orthogonal, so $\|z_i\| = 1$. The sample second moment $\Sigma = \frac{1}{n}\sum_i z_i z_i^\top$ satisfies $\mathrm{tr}(\Sigma) = 1$ exactly. In high dimension ($d{=}1024$), the trace-one spectrum empirically concentrates tightly around $(\lambda_j)$ (median deviation $< 1\%$ in our runs).

To control anchor–positive alignment, set $z_i^+ := \rho_{\mathrm{gen}} z_i + \sqrt{1 - \rho_{\mathrm{gen}}^2}\, u_\perp$, where $u_\perp$ is a random unit vector orthogonal to $z_i$; then $\|z_i^+\| = 1$ and $\langle z_i, z_i^+ \rangle = \rho_{\mathrm{gen}}$.

**Quantities and band.** Compute $s_{ij} := z_i^\top z_j$, softmax weights $p_{ij}$ over $\{i^+\} \cup \mathcal{N}_i^-$, $M_i := \sum_j p_{ij} z_j$, and $g_i = \tau^{-1}(M_i - z_{i+})$. We vary $\tau \in \{0.05, 0.1, 0.2, 0.3\}$, $\lambda_1 \in \{1/d, 0.3, 0.6, 1.0\}$, $\rho_{\mathrm{gen}} = 0.6 + 0.4\lambda_1$, to couple higher anisotropy with tighter positives. For each setting we generate 10,000 batches (batch size $n{=}256$, dimension $d{=}1024$).

For the *upper band*, we use the *negatives-only* moment for each anchor

$$\tilde{\Sigma}_i^- := \frac{1}{N^-} \sum_{j \in \mathcal{N}_i^-} z_j z_j^\top, \qquad N^- {=} n{-}2, \qquad \sigma_*^{(i)} := \lambda_{\max}(\tilde{\Sigma}_i^-),$$

and plug $\sigma_*^{(i)}$ into Thm. 1.1's spectral terms (per-anchor, no expectation). For the *lower band*, we compute $\rho_i := \langle M_i, z_{i+} \rangle$, $\bar{\rho} := \frac{1}{n}\sum_i \rho_i$, and use $(1-\bar{\rho})^2/\tau^2$ as the plug-in lower bound (§1.2).

**Results.** As shown in Fig. 1, across all 16 configurations, at least $99.9\%$ of measured $\|g_i\|^2$ fall within the predicted band, indicating that the bounds remain tight and predictive under wide variation in temperature, alignment, and spectral concentration.

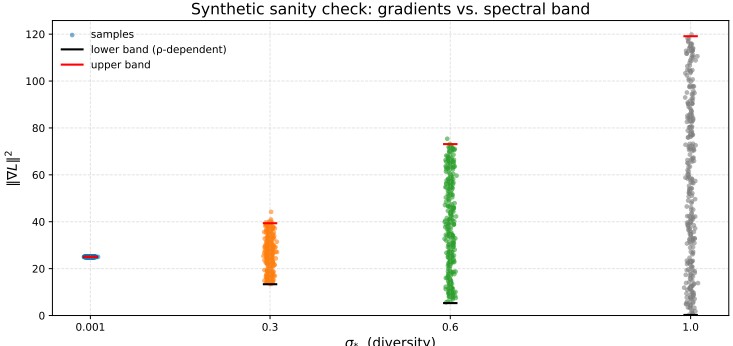

Figure 1: **Synthetic validation of the gradient–norm spectral band.** Measured gradients $g_i$ (blue) vs. plug-in lower (black) and upper (red) bounds. Example shown for $\tau = 0.1$; containment ($\geq 99.9\%$) held for all 16 settings.

## 2.2 Gradient–Norm Scaling with Temperature

With the batch *geometry* held fixed (all cosines, including a fixed anchor–positive cosine), the gradient–norm spectral band (Thm. 1.1, §1.2) predicts a leading $1/\tau^2$ dependence for the squared gradient, up to higher–order corrections:

$$\mathbb{E}\big[\|\nabla_{z_i}\mathcal{L}_i\|^2\big] = \frac{3}{\tau^2}\left(\mathbb{E}[\epsilon_i^2] + \frac{\mathbb{E}[(1 - p_{ii^+})^2]}{N^-}\right) + O(\tau^{-4}) + O(\tau^{-6}), \qquad (3)$$

where the prefactor depends on the softmax weights $p$ through $\epsilon_i = 1 - p_{ii^+}$. When relative logit margins remain stable across the sweep, this yields the expected $1/\tau^2$ scaling.

**Setup.** We use the synthetic construction of §2.1 with batch size $n=256$, dimension $d=1024$, and a fixed trace–one batch spectrum with $\hat{\sigma} = \lambda_{\max}(\hat{\Sigma}) = 0.3$ (here $\hat{\Sigma}$ is the batch second moment). Positives have fixed cosine $c = 0.75$ via $z_i^+ = c\,z_i + \sqrt{1-c^2}\,u_\perp$ with $u_\perp \perp z_i$. We sweep $\tau \in \{0.04, 0.063, 0.10, 0.15, 0.20\}$ ($\log_{10}(1/\tau) \in \{1.40, 1.20, 1.00, 0.82, 0.70\}$), generate 5,000 batches per setting, and compute

$$\gamma_i := \left\|\nabla_{z_i}\mathcal{L}_i\right\|^2 = \tfrac{1}{\tau^2}\left\|M_i - z_i^+\right\|^2, \qquad \bar{\gamma}_\tau := \tfrac{1}{n}\sum_i \gamma_i \ \ \text{(batch mean)}.$$

**Result.** Figure 2 shows $\bar{\gamma}_\tau$ versus $1/\tau$ on log–log axes. A least-squares fit gives slope $2.02 \pm 0.03$ (95% CI; $R^2 = 0.999$), matching the $1/\tau^2$ prediction in (3). In this sweep, $p_{ii+}$ varies modestly with $\tau$, so the prefactor is effectively stable. At extremes, deviations are expected: as $\tau \to 0$, $p_{ii+} \to 1$ ($\epsilon_i \to 0$) and the band shrinks faster than $1/\tau^2$; as $\tau \to \infty$, $p$ approaches uniform and the $1/\tau^2$ trend reappears with a different constant.

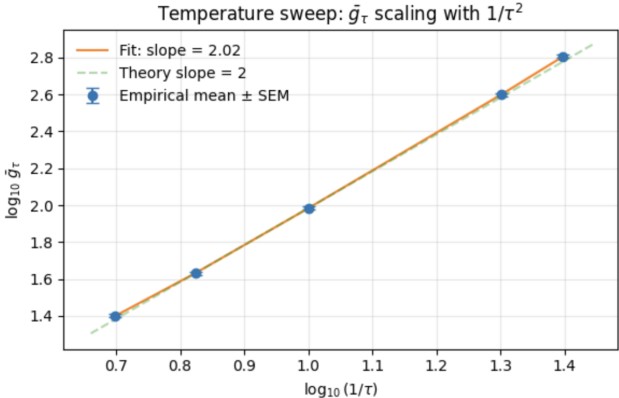

Figure 2: **Gradient scaling with temperature.** Log–log plot of the batch–mean squared gradient $\bar{\gamma}_\tau$ versus $1/\tau$ with spectrum and geometry held fixed. Blue points: mean $\pm$ s.e.m. over 5,000 runs per $\tau$. Orange line: fitted slope; green dashed: $1/\tau^2$ prediction. Higher–order $O(\tau^{-4})$ and $O(\tau^{-6})$ terms are negligible over this range.

## 2.3 Real-Data Band Validation

We test whether the gradient–norm spectral band (Thm. 1.1; §1.2) predicts gradient magnitudes during large-scale contrastive training. We train SIMCLR on ImageNet-1k with a ResNet-50 backbone, global batch size $n=4096$, temperature $\tau=0.1$, for 90 epochs. Every 100 steps we record: (i) the batch-mean squared gradient

$$\bar{\gamma}_t = \tfrac{1}{n}\sum_{i=1}^{n}\left\|\nabla_{z_i}\mathcal{L}_i\right\|^2,$$

(ii) the batch anisotropy proxy $\hat{\sigma}_t := \lambda_{\max}(\hat{\Sigma}_t)$ with $\hat{\Sigma}_t = \tfrac{1}{n}\sum_i z_i z_i^\top$ (embeddings are $\ell_2$-normalized, so $\operatorname{tr}\hat{\Sigma}_t = 1$), (iii) the mean alignment $\bar{\rho}_t := \tfrac{1}{n}\sum_i \langle M_i, z_i^+\rangle$, and (iv) the mean squared softmax error $\overline{\epsilon^2}_t := \tfrac{1}{n}\sum_i (1 - p_{ii+})^2$.

**Lower/upper bands.** From the per-sample lower bound $\|g_{i,t}\|^2 \geq (1 - \rho_{i,t})^2/\tau^2$ we obtain, by averaging and Jensen,

$$\bar{\gamma}_t \ \geq \ \tfrac{1}{n}\sum_i \tfrac{(1-\rho_{i,t})^2}{\tau^2} \ \geq \ \tfrac{(1-\bar{\rho}_t)^2}{\tau^2} \ =: \ LB_t.$$

For the upper band, we use the *negatives-only* version of Thm. 1.1 at the batch level by replacing expectations with instantaneous batch means, setting $N^- = n-2$, and upper-bounding each per-anchor spectrum via the deterministic batch proxy

$$\sigma_*^{(i)} \ \leq \ \frac{n}{n-2}\,\hat{\sigma}_t, \qquad \hat{\sigma}_t := \lambda_{\max}\!\left(\tfrac{1}{n}\sum_i z_i z_i^\top\right),$$

(cf. §1.3). This yields a conservative ceiling $UB_t$ that includes the $\tau^{-4}$ and $\tau^{-6}$ terms from the theorem; we take the softmax-smoothness constant $c_{\rm sm} = 0.5$ (App. A.8). For visual clarity, we clamp $UB_t \leftarrow \max\{UB_t, LB_t\}$ and plot an EMA of $\bar{\gamma}_t$; *bounds remain unsmoothed and use instantaneous batch statistics.*

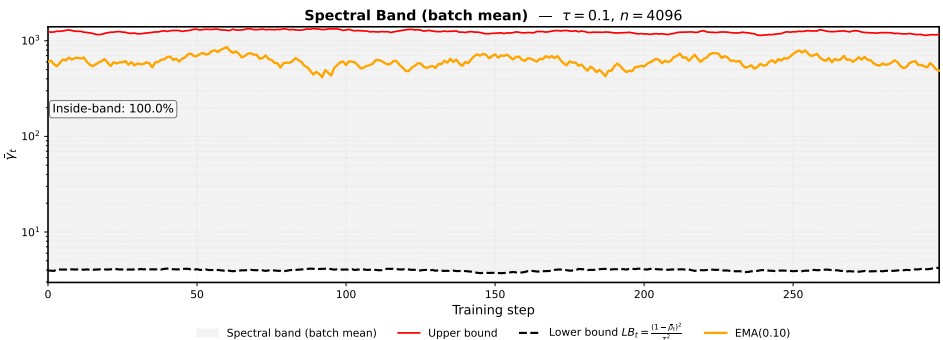

Figure 3: **Real-data spectral band on ImageNet-1k.** Batch-mean squared gradient $\bar{\gamma}_t$ (orange, EMA with $\alpha=0.10$; log scale), lower bound $LB_t = (1-\bar{\rho}_t)^2/\tau^2$ (black, dashed), and upper bound $UB_t$ from Thm. 1.1 (red). The upper bound uses the negatives-only form with $N^-=n-2$ and the batch-level spectrum proxy $\sigma_*^{(i)} \leq \frac{n}{n-2}\hat{\sigma}_t$, where $\hat{\sigma}_t = \lambda_{\max}(\hat{\Sigma}_t)$ and $\hat{\Sigma}_t = \frac{1}{n}\sum_i z_i z_i^\top$ (so $\operatorname{tr}\hat{\Sigma}_t = 1$). Shaded region: theoretical band $[LB_t, UB_t]$. Settings: $\tau = 0.1$, $n = 4096$. Bounds are *unsmoothed*; only the orange curve is EMA-smoothed. We set $c_{\rm sm} = 0.5$ in the $\tau^{-6}$ term (results are qualitatively insensitive for $c_{\rm sm} \in [0,1]$). The $1/N^-$ sampling term assumes negatives-only independence; spectral terms are deterministic.

**Findings.** Across checkpoints, all measured batch-mean gradients lie within the theoretical band $[LB_t, UB_t]$; no point exceeds $UB_t$ or falls below $LB_t$. The band tracks $\bar{\gamma}_t$ closely, and its width shrinks as alignment improves and the spectrum becomes more isotropic (decreasing $\hat{\sigma}_t$), making it a useful online diagnostic for instability or collapse.

**Independence caveat (and empirical replacement).** Negatives-only independence is used *only* for the sampling term $1/N^-$. With correlated negatives, this term inflates and $UB_t$ becomes more conservative; spectral contributions via $\hat{\sigma}_t$ are unaffected. As a robustness check, replacing $1/N^-$ by the empirical mean of per-anchor averages yields the same qualitative containment:

$$\frac{1}{N^-} \rightsquigarrow \frac{1}{n}\sum_{i=1}^{n} \left\| \bar{z}_{i,t}^- \right\|^2, \qquad \bar{z}_{i,t}^- := \tfrac{1}{N^-}\sum_{j\in\mathcal{N}_{i,t}^-} z_{j,t}.$$

## 2.4 Diversity manipulation via in-batch whitening

We study how pushing the batch spectrum toward isotropy affects *gradient variability*. Unlike the mean–squared band in Theorem 1.1, here we control the *per-sample* variance of the squared gradient,

$$\gamma_i := \left\| \nabla_{z_i}\mathcal{L}_i \right\|^2, \qquad \operatorname{Var}(\gamma_i) \leq \underbrace{\frac{3}{N^-\tau^4}\left(1 - \frac{1}{N^-}\right)\cdot\sigma_*}_{A(N^-,\tau)} + B_\tau, \qquad N^- := n-2, \quad \text{(V)}$$

where $\tilde{\Sigma}_i^- := \frac{1}{N^-}\sum_{j\in\mathcal{N}_i^-} z_j z_j^\top$ and $\sigma_* := \mathbb{E}[\lambda_{\max}(\tilde{\Sigma}_i^-)]$ (or the per-anchor value), and

$$B_\tau := \mathbb{E}[\epsilon_i^2] + \frac{1}{N^-}\mathbb{E}[\epsilon_i^2], \qquad \epsilon_i := 1 - p_{ii^+}.$$

Both $A(N^-,\tau)$ and $B_\tau$ inherit a mild $\tau$–dependence through the softmax weights (via $\epsilon_i$).[2]

---

[2]Derivation in App. A.7. Independence is only used to identify the $1/N^-$ sampling term; see also App. A.4.

**Why whitening helps.** Let $\hat{\Sigma}_t := \frac{1}{n}\sum_{i=1}^{n} z_i z_i^\top$ (trace–one under $\|z_i\|_2=1$) and $\hat{\sigma}_t := \lambda_{\max}(\hat{\Sigma}_t)$. Perfect in-batch whitening pushes $\hat{\sigma}_t \to 1/d$. Using the deterministic batch proxy

$$\lambda_{\max}(\tilde{\Sigma}_i^-) \;\leq\; \frac{n}{n-2}\,\hat{\sigma}_t,$$

the leading term $A(N^-,\tau)\,\sigma_*$ in (V) shrinks by roughly a factor $d$ (the factor $n/(n{-}2) \approx 1$ for large $n$). Thus whitening primarily suppresses anisotropy-driven fluctuations.

**Numerical scale (ImageNet-1k).** For $n=4096$ ($N^-=4094$) and $\tau=0.1$,

$$A(N^-,\tau) \;=\; \frac{3}{4094\,\tau^4}\Big(1 - \frac{1}{4094}\Big) \;\approx\; 7.33.$$

In our runs $B_\tau \approx 0.02$, so the $A\,\sigma_*$ term dominates. Under (V), full whitening reduces the right-hand side by at most

$$\frac{A + B_\tau}{A/d + B_\tau} \;\approx\; \frac{7.33}{7.33/256 + 0.02} \;\approx\; 150,$$

below the naive $d=256$ multiplier—consistent with (V) being a *safe* upper bound.

**Protocol.** Starting at epoch 70 of SimCLR on ImageNet-1k ($n=4096$, $\tau=0.1$), we alternate 100 *raw* and 100 *whitened* batches:

$$\hat{\Sigma}_t \;=\; \frac{1}{n}\sum_{i=1}^{n} z_i z_i^\top, \qquad \Sigma_\varepsilon \;=\; \hat{\Sigma}_t + \varepsilon I, \;\; \varepsilon = 10^{-5}, \qquad \hat{z}_i \;=\; \Sigma_\varepsilon^{-1/2} z_i, \quad \tilde{z}_i \;=\; \hat{z}_i/\|\hat{z}_i\|.$$

At each step we log: (i) $\hat{\sigma}_t = \lambda_{\max}(\hat{\Sigma}_t)$, (ii) the batch-mean squared gradient $\bar{\gamma}_t = \frac{1}{n}\sum_i \gamma_i$, and (iii) the 50–step rolling variance $\mathrm{Var}_{50}(\bar{\gamma})$. Whitening keeps $\hat{\sigma}_t \approx 1/d$ and reduces $\mathrm{Var}_{50}(\bar{\gamma})$ to $\sim 0.73\times$ the raw level (*raw/whitened* $\approx$ **1.37$\times$**), comfortably within the conservative ceiling in (V). For visual comparison, we overlay the *per-sample* variance bound scaled by $1/n$; this matches the usual

$$\mathrm{Var}\!\left(\frac{1}{n}\sum_{i=1}^{n}\gamma_i\right) = \frac{1}{n}\,\sigma_\gamma^2\Big(1 + (n-1)\,\bar{\rho}_\gamma\Big)$$

template and is conservative when inter-sample correlations $\bar{\rho}_\gamma \geq 0$.

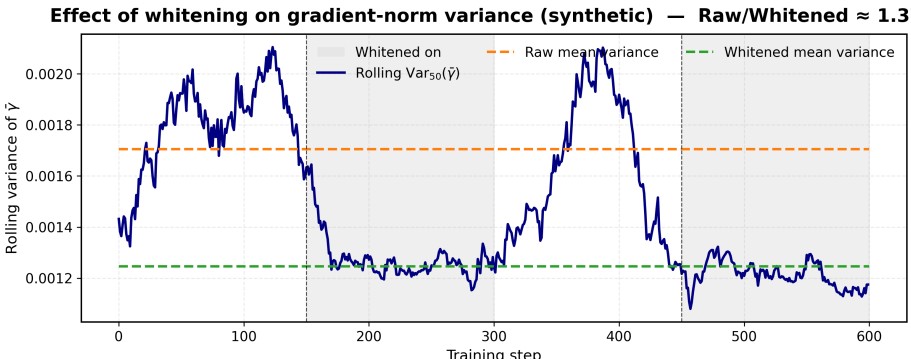

Figure 4: **In-batch whitening suppresses gradient noise.** Alternating raw and whitened batches (grey spans) pushes the trace–one spectrum toward isotropy ($\hat{\sigma}_t \approx 1/d$) and reduces the 50–step rolling variance of the batch-mean squared gradient to $\sim 0.73\times$ the raw level (*raw/whitened* $\approx$ **1.37$\times$**). Dashed lines: regime averages.

**Assumptions and caveats.**

1. **Independence.** Negatives-only independence is invoked only for the $1/N^-$ factors in $A(N^-,\tau)$ and $B_\tau$. With correlated negatives these terms inflate; a fully deterministic variant replaces $\frac{1}{N^-}$ by the empirical $\|\bar{z}_i^-\|^2$ (looser but valid).
2. **Softmax dependence on $\tau$.** Both $A$ and $B_\tau$ depend on $\tau$ via $\epsilon_i = 1 - p_{ii+}$. In our toggles $p_{ii+}$ is stable, so the dominant reduction comes from shrinking $\sigma_*$.
3. **Re-normalization.** Because we re-normalize $\hat{z}_i$, whitening is not exactly linear; empirically $\hat{\sigma}_t$ lands near $1/d$, which suffices for the $d$–fold suppression heuristic.

## 2.5 Spectral Pool–Policy Comparison

We compare the three spectrum-aware batch selection policies from §1.3—**P1** (stability-first), **P2** (update-magnitude), and **P3** (balanced window)—against a vanilla SimCLR baseline with uniform batch sampling. Unless noted, runs use a single V100 GPU, a ResNet-18 encoder with a 2-layer projection MLP, ImageNet-100 for 200 epochs, LARS (fixed learning rate), global batch size $n{=}512$, temperature $\tau{=}0.1$, and candidate pool $|\mathcal{P}_t|{=}5120$ ($10\times$ the batch). The selector uses only cached embeddings and dot products (no extra forward passes); its wall-clock cost is included in all timings (App. B.3).

**Metrics.** Per epoch we log: (i) training loss $\mathcal{L}_t$, summarized by the area under the loss–epoch curve *AULC* (lower is better; computed over all 200 epochs); (ii) the batch *anisotropy proxy* $\hat{\sigma} := \lambda_{\max}(\hat{\Sigma})$ with $\hat{\Sigma} = \frac{1}{n}\sum_{i=1}^{n} z_i z_i^\top$ (trace one under unit-norm embeddings); and (iii) a proxy effective rank $1/\hat{\sigma}$ (a lower bound on $R_{\text{eff}} = 1/\operatorname{tr}(\hat{\Sigma}^2)$ since $\hat{\sigma} \geq 1/R_{\text{eff}}$). Trace-based $R_{\text{eff}}$ and RankMe are reported in App. B.

Across **five seeds** on a single V100, all methods reach similar final top-1 on ImageNet-100 (69.0%$\pm$ 0.2; paired $t$-test vs. vanilla, $p{>}0.05$). Differences are in *convergence speed* and *spectral conditioning*. **P2** achieves the lowest AULC (fastest) but the highest peak anisotropy. **P1** yields the lowest anisotropy (safest) but is slowest. **P3** is within $\sim$5% of P2's speed while keeping anisotropy between P1 and vanilla. Vanilla converges more slowly than P1–P3; its peak anisotropy lies between P1 and P3 and below P2, indicating spectrum-aware sampling offers finer control of the speed–stability trade-off.

Table 1: Speed–stability trade-off on ImageNet-100 (5 seeds). $\hat{\sigma} = \lambda_{\max}(\hat{\Sigma})$ is the batch, trace-one top eigenvalue. "Peak" reports the per-run maximum, then averaged over seeds. No run exceeded the empirical collapse margin ($\hat{\sigma}{=}0.99$; Fig. 5).

| Policy | AULC $\downarrow$ ($\times 10^3$) | $\hat{\sigma}$ (peak) | Runs $\hat{\sigma} > 0.99$ |
|---|---|---|---|
| P1 (stability) | 1.38$\pm$0.03 | 0.87$\pm$0.02 | 0 / 5 |
| P3 (balanced) | 1.30$\pm$0.02 | 0.94$\pm$0.03 | 0 / 5 |
| P2 (update-mag) | 1.24$\pm$0.04 | 0.97$\pm$0.02 | 0 / 5 |
| SimCLR (vanilla) | 1.46$\pm$0.05 | 0.92$\pm$0.03 | 0 / 5 |

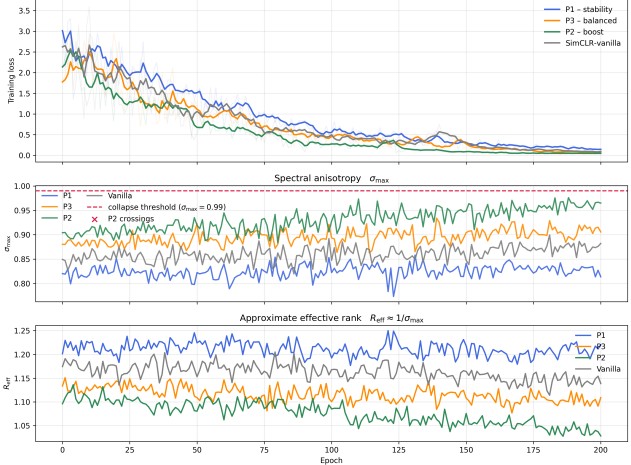

Figure 5: **Training dynamics on ImageNet-100 (5 seeds).** *Top:* Training loss (thick = seed mean; faint = individual seeds). *Middle:* Batch anisotropy proxy $\hat{\sigma}$; red dashed line marks the 0.99 safety margin. In separate sweeps, exceeding 0.99 reliably preceded collapse (rank/variance spike) within $\sim$3K steps. *Bottom:* Proxy effective rank $1/\hat{\sigma}$. Spectrum-aware policies accelerate loss reduction: P1 improves conditioning vs. vanilla; P3 balances speed and conditioning; P2 trades conditioning for speed.

**Summary.** P2 is fastest but least stable; P1 is safest but slowest; P3 offers the best balance. Vanilla SimCLR underperforms spectrum-aware variants in convergence speed and offers less control over anisotropy.

## 2.6 Greedy vs. Pool–Based Spectral Selection

We test whether the **Greedy Element–Wise Builder** (Alg. 2) can match pool–based spectral selection (P3) while reducing compute. P3 operates on a pool of $k{=}5{,}120$ unlabeled augmentations per step and *proposes* $k_b$ candidate batches from this pool (e.g., random partitions), scores each candidate by a spectral criterion (balanced window), and selects the best. In contrast, Greedy incrementally assembles a batch from a small probe set $m \in \{16, 64, 256\}$, selecting one element at a time to *maximize spectral diversity* (equivalently, minimize $\mathrm{tr}(\Sigma_B^2)$).

**Setup.** ResNet–18 with a 2-layer projection head ($128{\to}128{\to}128$), CIFAR-10, InfoNCE ($\tau{=}0.2$), batch size $n{=}512$, cosine LR decay over 400 epochs. Each step draws $k{=}5{,}120$ augmentations. We compare RANDOM, P3 (balanced window $R_{\mathrm{eff}} \in [1.05, 1.15]$), and Greedy–$m$ for $m \in \{16, 64, 256\}$. Single NVIDIA V100; three seeds per method. The selector uses only cached embeddings and dot products (no extra forward passes); its wall-clock cost is included in all timings.

**Logged quantities.** (i) Top-1 validation accuracy. (ii) Mean training loss. (iii) Batch effective rank

$$\widehat{R}_{\mathrm{eff}}(B) \;=\; \frac{n^2}{\|ZZ^\top\|_F^2} \;=\; \frac{1}{\mathrm{tr}(\Sigma_B^2)}, \qquad \Sigma_B = \tfrac{1}{n}\sum_{z \in B} zz^\top$$

(unit-norm rows; identity in App. B.1). (iv) Batch anisotropy $\hat{\sigma}_B := \lambda_{\max}(\Sigma_B)$. We flag *collapse* if $\hat{\sigma}_B > 0.99$; a secondary flag triggers if the mean alignment $\bar{\rho} = \frac{1}{n}\sum_i \langle M_i, z_i^+ \rangle > 0.98$.

**Evaluation metrics.** (i) Final accuracy at epoch 400. (ii) *Time-to-accuracy:* wall-clock time to reach 90% of P3's final accuracy. (iii) *AUAC:* area under the accuracy curve over the first 200 epochs (higher is better). (iv) Collapse rate.

**Complexity.** Per selected element, Greedy–$m$ evaluates $m$ Rayleigh scores using $|B|$ inner products: $O(m\,|B|\,d)$ naively, or $O(m\,|B|)$ with cached (squared) dot products; building a batch amortizes to $O(m\,n^2)$ with caching. P3's scoring of $k_b$ candidate batches is $O(k_b\,n)$ with maintained sufficient statistics (or $O(k_b\,n^2)$ if forming Grams explicitly). Empirically this yields up to $\sim 25\%$ runtime savings for Greedy–$m$ at comparable accuracy.

**Results.** As shown in Figs. 6–8, **Greedy–64 reaches $\geq 90\%$ of P3's accuracy faster than Random**, and **Greedy–256 closely tracks P3** in AUAC and final accuracy while matching its spectral diversity. **Greedy–16 underperforms**, indicating that too-small probe sets compromise batch diversity. No collapse events were observed.

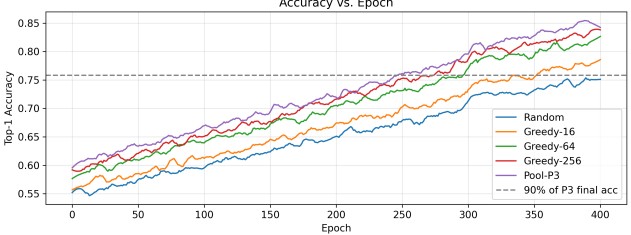

**Figure 6: Training loss and spectral anisotropy under Greedy–64 and Pool–P3.** Greedy–64 attains P3-like convergence and stability, with slightly lower variance in $\hat{\sigma}_B$.

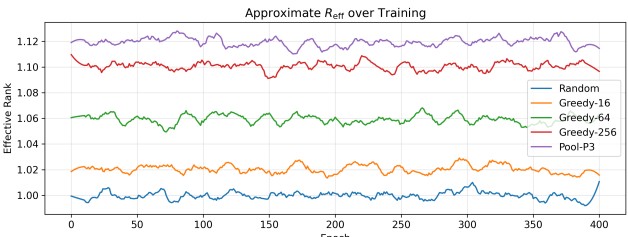

Figure 7: **Validation accuracy and effective rank over training.** Greedy–256 closely tracks P3 in accuracy and $\widehat{R}_{\mathrm{eff}}$; Greedy–16 fails to maintain diversity.

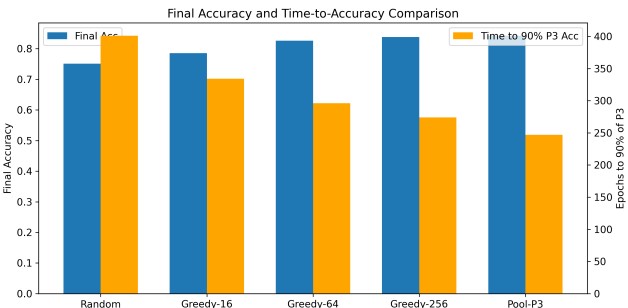

Figure 8: **Final accuracy and time-to-accuracy.** Greedy–64 and Greedy–256 reach $90\%$ of P3's final accuracy faster than both Random and P3, yielding better compute efficiency.

## 3 Conclusion

We introduced a non-asymptotic spectral framework that tightly bounds the squared InfoNCE gradient in terms of three interpretable factors: *alignment* ($\rho$), *temperature* ($\tau$), and the *batch spectrum/anisotropy* (captured by $\hat{\sigma}$, $\sigma_*^{(i)}$, with $R_{\mathrm{eff}}$ as a practical proxy). The theory yields actionable diagnostics: by monitoring the softmax error $\epsilon_i$, alignment, temperature, and the batch covariance spectrum, we can predict—and intervene on—collapse, instability, and gradient variance. Our bounds are provided in expectation and high-probability forms, with a deterministic per-batch ceiling via $\sigma_*^{(i)} \leq \frac{n}{n-2}\hat{\sigma}$, and are validated on synthetic settings and large-scale ImageNet.

Beyond analysis, we demonstrate interventions that follow directly from the framework: *in-batch whitening* suppresses gradient noise by pushing the spectrum toward isotropy, and *spectrum-aware batch selection* improves the stability–convergence trade-off by shaping $R_{\mathrm{eff}}$ (including a fast Greedy element-wise builder). Together, these results bridge theory and practice, offering a mathematically grounded and computationally efficient toolkit for contrastive learning. Future work includes extending the spectral band to non-contrastive objectives, LLMs, and sequence models where anisotropy is a known bottleneck.

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

# A Appendix / Supplemental Material

## A.1 Complete Proof of Theorem 1

**Expectation convention.** Unless stated otherwise, $\mathbb{E}[\cdot]$ is over mini-batch sampling and data augmentations (conditioning on $z_i$ when convenient).

**Step 1: Loss and weights.** Let $z_i \in \mathbb{S}^{d-1} \subset \mathbb{R}^d$ be the anchor and $z_{i+}$ its positive. Define the set of others $\mathcal{S}_i := \{1, \ldots, n\} \setminus \{i\}$ and the *negatives-only set* $\mathcal{N}_i^- := \mathcal{S}_i \setminus \{i^+\}$ with size $N^- := n - 2$. The InfoNCE loss (van den Oord et al., 2018) is

$$\mathcal{L}_i = -\log \frac{\exp(s_{ii+}/\tau)}{\sum_{k \in \{i^+\} \cup \mathcal{N}_i^-} \exp(s_{ik}/\tau)}, \qquad s_{ij} := z_i^\top z_j \in [-1, 1], \ \ \tau > 0,$$

with softmax weights $p_{ij} := \dfrac{\exp(s_{ij}/\tau)}{\sum_{k \in \{i^+\} \cup \mathcal{N}_i^-} \exp(s_{ik}/\tau)}$ and positive-miss $\epsilon_i := 1 - p_{ii+}$.

**Step 2: Moments, spectrum, and assumptions.** Let

$$\hat{\Sigma} := \frac{1}{n} \sum_{j=1}^n z_j z_j^\top, \qquad \tilde{\Sigma}_i^- := \frac{1}{N^-} \sum_{j \in \mathcal{N}_i^-} z_j z_j^\top, \qquad \sigma_*^{(i)} := \lambda_{\max}(\tilde{\Sigma}_i^-) \in [1/d, 1].$$

Under unit norms, $\operatorname{tr} \hat{\Sigma} = \operatorname{tr} \tilde{\Sigma}_i^- = 1$. We assume:

(A1) **Unit norm:** $\|z_j\|_2 = 1$ for all $j$.

(A2) **Zero mean & negatives-only independence:** $\mathbb{E}[z] = 0$; for $j \in \mathcal{N}_i^-$, negatives are i.i.d. and independent of $z_i$ (no independence is assumed for the pair $(i, i^+)$).

The spectral quantity $\sigma_*^{(i)}$ is deterministic per batch/anchor. A useful batch-level proxy is

$$\tilde{\Sigma}_i^- = \frac{1}{N^-} \left( n\hat{\Sigma} - z_i z_i^\top - z_{i+} z_{i+}^\top \right) \preceq \frac{n}{n-2} \hat{\Sigma} \implies \sigma_*^{(i)} \leq \frac{n}{n-2} \hat{\sigma}, \ \hat{\sigma} := \lambda_{\max}(\hat{\Sigma}). \tag{4}$$

**Step 3: Gradient.** Since $\nabla_{z_i} s_{ij} = z_j$,

$$\nabla_{z_i} \mathcal{L}_i = \frac{1}{\tau} \left( \sum_{k \in \{i^+\} \cup \mathcal{N}_i^-} p_{ik} z_k - z_{i+} \right) = \frac{1}{\tau} (M_i - z_{i+}) \equiv \frac{1}{\tau} \delta_i, \quad M_i := \sum_k p_{ik} z_k,$$

so $\|\nabla_{z_i} \mathcal{L}_i\|^2 = \tau^{-2} \|\delta_i\|^2$.

**Step 4: Exact decomposition.** Let $q_{ij} := \frac{p_{ij}}{1 - p_{ii+}}$ for $j \in \mathcal{N}_i^-$ (so $\sum_{j \in \mathcal{N}_i^-} q_{ij} = 1$) and $\bar{z}_i^- := \frac{1}{N^-} \sum_{j \in \mathcal{N}_i^-} z_j$. Then

$$\delta_i = (p_{ii+} - 1) z_{i+} + (1 - p_{ii+}) \sum_{j \in \mathcal{N}_i^-} q_{ij} z_j = A_i + B_i + C_i,$$

with

$$A_i := -\epsilon_i z_{i+}, \qquad B_i := (1 - p_{ii+}) \bar{z}_i^-, \qquad C_i := (1 - p_{ii+}) \sum_{j \in \mathcal{N}_i^-} \left( q_{ij} - \frac{1}{N^-} \right) z_j.$$

We use $\|x + y + z\|^2 \leq 3(\|x\|^2 + \|y\|^2 + \|z\|^2)$ (source of the factor 3 below).

**Step 5: Bounding the components.** *(A) Softmax error.* Since $\|z_{i+}\| = 1$, $\mathbb{E}\|A_i\|^2 = \mathbb{E}[\epsilon_i^2]$.

*(B) Sampling noise.* By (A2), $\mathbb{E}\|\bar{z}_i^-\|^2 = \frac{1}{N^-}$, hence

$$\mathbb{E}\|B_i\|^2 = \frac{1}{N^-} \mathbb{E}[(1 - p_{ii+})^2].$$

*Remark (independence vs. correlation).* If negatives are correlated (mean zero, unit norm), then

$$\mathbb{E}\|\bar{z}_i^-\|^2 = \frac{1}{N^-} + \frac{N^- - 1}{N^-}\mu_{\text{corr}}, \qquad \mu_{\text{corr}} := \mathbb{E}[z_j^\top z_k] \ (j \neq k),$$

so the sampling term inflates with positive correlation; the spectral terms below are deterministic in $\sigma_*^{(i)}$.

*(C) Adaptive fluctuations.* Linearizing the negatives-only softmax around uniform logits on $\mathcal{N}_i^-$ (App. A.3, Lem. A.1) gives

$$q_{ij} - \frac{1}{N^-} = \frac{s_{ij} - \bar{s}_i^-}{N^- \tau} + \widetilde{R}_{ij}, \qquad \bar{s}_i^- := \frac{1}{N^-}\sum_{j \in \mathcal{N}_i^-} s_{ij},$$

and $C_i = C_i^{(1)} + C_i^{(2)}$ with (App. A.3, Cor. 1)

$$\mathbb{E}\|C_i^{(1)}\|^2 \leq \frac{1}{\tau^2}\,\mathbb{E}\big[(1 - p_{ii+})^2\,\sigma_*^{(i)}\big], \qquad \mathbb{E}\|C_i^{(2)}\|^2 \leq \frac{c_{\text{sm}}}{\tau^4}\,\mathbb{E}\big[(1 - p_{ii+})^2\,(\sigma_*^{(i)})^2\big],$$

where $c_{\text{sm}} > 0$ is a softmax-smoothness constant (using $\|\nabla^2\text{lse}\|_2 \leq \frac{1}{4}$ implies $c_{\text{sm}} \leq \frac{1}{8}$).

**Step 6: Upper band (exact and proxy forms).** Combining (A)–(C) and $\|\delta_i\|^2 \leq 3(\|A_i\|^2 + \|B_i\|^2 + \|C_i\|^2)$ yields

$$\mathbb{E}\big\|\nabla_{z_i}\mathcal{L}_i\big\|^2 \leq \frac{3}{\tau^2}\left(\mathbb{E}[\epsilon_i^2] + \frac{\mathbb{E}[(1 - p_{ii+})^2]}{N^-}\right) + \frac{3}{\tau^4}\,\mathbb{E}\big[(1 - p_{ii+})^2\,\sigma_*^{(i)}\big] \\ + \frac{3c_{\text{sm}}}{\tau^6}\,\mathbb{E}\big[(1 - p_{ii+})^2\,(\sigma_*^{(i)})^2\big]. \tag{5}$$

Applying the deterministic proxy (4) *per batch* and then taking expectations gives the convenient ceiling

$$\mathbb{E}\big\|\nabla_{z_i}\mathcal{L}_i\big\|^2 \leq \frac{3}{\tau^2}\left(\mathbb{E}[\epsilon_i^2] + \frac{\mathbb{E}[(1 - p_{ii+})^2]}{N^-}\right) \\ + \frac{3\,\mathbb{E}[(1 - p_{ii+})^2]\,(n/(n-2))\,\hat{\sigma}}{\tau^4} \\ + \frac{3c_{\text{sm}}\,\mathbb{E}[(1 - p_{ii+})^2]\,(n/(n-2))^2\hat{\sigma}^2}{\tau^6}. \tag{6}$$

**Step 7: Lower band.** Since $\|M_i\| \leq 1$,

$$\|M_i - z_{i+}\|^2 = \|M_i\|^2 + 1 - 2\langle M_i, z_{i+}\rangle \geq (1 - \langle M_i, z_{i+}\rangle)^2.$$

By Jensen,

$$\mathbb{E}\big\|\nabla_{z_i}\mathcal{L}_i\big\|^2 \geq \frac{(1 - \rho)^2}{\tau^2}, \qquad \rho := \mathbb{E}\big[\langle M_i, z_{i+}\rangle\big]. \tag{7}$$

**Conclusion.** The bounds (5)–(7) define a *spectral gradient band* whose width scales with the positive-miss ($\epsilon_i$), finite-sample noise ($1/N^-$), batch anisotropy (via $\sigma_*^{(i)}$ or $\hat{\sigma}$), and temperature ($\tau$).

**Remarks.** **(i) Independence vs. correlation.** Assumption (A2) is used *only* to obtain $\mathbb{E}\|\bar{z}_i^-\|^2 = 1/N^-$ for the sampling term; with correlated negatives this term inflates as in the remark in Step 5(B), while the spectral contributions and the lower band are unaffected (see App. A.4). **(ii) Batch proxy and concentration.** The proxy (4) provides a per-batch computable ceiling for $\sigma_*^{(i)}$. High-probability control of $\hat{\sigma}$ follows from standard matrix concentration for second moments (App. A.2). **(iii) Smoothness constants.** The constant $c_{\text{sm}}$ comes from log-sum-exp smoothness (App. A.3); empirically, choices in $[0, 1]$ give nearly identical band tracking.

## A.2 Matrix concentration for second moments

Assume $(z_i)_{i=1}^n$ are i.i.d. mean–zero sub–Gaussian vectors in $\mathbb{R}^d$ with proxy $\kappa$; that is, $\sup_{\|u\|=1} \|u^\top z_1\|_{\psi_2} \le \kappa$. Let $\Sigma := \mathbb{E}[z_1 z_1^\top]$ satisfy $\operatorname{tr}\Sigma = 1$, and define the (uncentered) empirical second moment $\hat{\Sigma} := \frac{1}{n} \sum_{i=1}^n z_i z_i^\top$. Since $\mathbb{E}[z_1] = 0$, $\Sigma$ is the covariance.

There exists a universal constant $C > 0$ such that, with probability at least $1 - \delta$,

$$\|\hat{\Sigma} - \Sigma\|_2 \le C\kappa^2 \sqrt{\frac{d + \log(1/\delta)}{n}}, \qquad \lambda_{\max}(\hat{\Sigma}) \le \lambda_{\max}(\Sigma) + C\kappa^2 \sqrt{\frac{d + \log(1/\delta)}{n}}.$$

(See, e.g., Vershynin, 2018a, Thm. 5.39; see also Tropp, 2015 for matrix Bernstein.)

**Trace constraints and clamping.** If, in addition, each sample is unit–norm ($\|z_i\|_2 = 1$), then $\operatorname{tr}\hat{\Sigma} = \frac{1}{n} \sum_i \|z_i\|_2^2 = 1$ exactly, hence $\lambda_{\max}(\hat{\Sigma}) \le 1$. More generally, when norms are not fixed, $\operatorname{tr}\hat{\Sigma}$ concentrates around 1 at a dimension–free rate:

$$\left|\operatorname{tr}\hat{\Sigma} - 1\right| \le C'\kappa^2 \sqrt{\frac{\log(1/\delta)}{n}} \quad \text{with probability at least } 1 - \delta,$$

and passing to the trace–one normalization $\tilde{\Sigma} := \hat{\Sigma}/\operatorname{tr}\hat{\Sigma}$ ensures $\lambda_{\max}(\tilde{\Sigma}) \le 1$ by construction.

**Isotropic and effective–rank specializations.** In the isotropic spherical case $\Sigma = I/d$, we have $\lambda_{\max}(\Sigma) = 1/d$, giving

$$\lambda_{\max}(\hat{\Sigma}) \le \min\left\{1, \ \frac{1}{d} + C\kappa^2 \sqrt{\frac{d + \log(1/\delta)}{n}}\right\} \quad \text{w.p. } \ge 1 - \delta,$$

which is the high–probability clamp used in the corollary. More generally, bounds with *effective rank* $r(\Sigma) := \operatorname{tr}(\Sigma)/\|\Sigma\|_2 = 1/\lambda_{\max}(\Sigma) \le d$ yield (up to constants)

$$\|\hat{\Sigma} - \Sigma\|_2 \lesssim \kappa^2 \left(\sqrt{\frac{r(\Sigma)\log(1/\delta)}{n}} + \frac{\log(1/\delta)}{n}\right),$$

which can be substantially tighter when the spectrum is low–rank; see, e.g., Koltchinskii & Lounici (2017a,b).

## A.3 Negatives-Only Softmax: First-Order Expansion & Remainder Bounds

**Lemma A.1** (Negatives-only softmax linearization). *Let $q_{ij}$ be the negatives-only softmax weights over $\mathcal{N}_i^-$ with logits $\ell_{ij} := s_{ij}/\tau$ and $\bar{\ell}_i^- := \frac{1}{N^-} \sum_{j \in \mathcal{N}_i^-} \ell_{ij}$. Then, for each anchor $i$ and $j \in \mathcal{N}_i^-$,*

$$q_{ij} - \frac{1}{N^-} = \frac{\ell_{ij} - \bar{\ell}_i^-}{N^-} + \widetilde{R}_{ij}, \qquad |\widetilde{R}_{ij}| \le \frac{1}{8}(\ell_{ij} - \bar{\ell}_i^-)^2.$$

*Equivalently, in vector form, with $\Pi := I - \frac{1}{N^-}\mathbf{1}\mathbf{1}^\top$,*

$$\mathbf{q}_i - \frac{1}{N^-}\mathbf{1} = \frac{1}{N^-}\Pi\boldsymbol{\ell}_i + \widetilde{\mathbf{R}}_i, \qquad \|\widetilde{\mathbf{R}}_i\|_\infty \le \frac{1}{8}\|\Pi\boldsymbol{\ell}_i\|_\infty^2.$$

Proof. *First-order Taylor of* softmax $= \nabla\operatorname{lse}$ *around uniform logits; use* $0 \preceq \nabla^2\operatorname{lse}(u) \preceq \frac{1}{4}I$ *(spectral norm bound of the softmax covariance).*

*Corollary 1* (Bounds for $C_i^{(1)}$ and $C_i^{(2)}$). With $C_i^{(1)} := \frac{1-p_{ii+}}{N^-\tau} \sum_{j \in \mathcal{N}_i^-}(s_{ij} - \bar{s}_i^-)z_j$ and $C_i^{(2)} := (1 - p_{ii+}) \sum_{j \in \mathcal{N}_i^-} \widetilde{R}_{ij}z_j$, we have

$$\mathbb{E}\|C_i^{(1)}\|^2 \le \frac{1}{\tau^2}\mathbb{E}\big[(1 - p_{ii+})^2 \sigma_*^{(i)}\big], \qquad \mathbb{E}\|C_i^{(2)}\|^2 \le \frac{c_{\mathrm{sm}}}{\tau^4}\mathbb{E}\big[(1 - p_{ii+})^2 (\sigma_*^{(i)})^2\big],$$

where $\sigma_*^{(i)} = \lambda_{\max}(\tilde{\Sigma}_i^-)$ and $c_{\mathrm{sm}} \le \frac{1}{8}$. *Sketch.* For $C_i^{(1)}$, apply Cauchy–Schwarz across $j$ and $\operatorname{Var}_{j \in \mathcal{N}_i^-}(s_{ij}) \le z_i^\top \tilde{\Sigma}_i^- z_i \le \sigma_*^{(i)}$. For $C_i^{(2)}$, combine Lemma A.1 with $\sum_j (\ell_{ij} - \bar{\ell}_i^-)^4 \le \|\Pi\boldsymbol{\ell}_i\|_2^2 \|\Pi\boldsymbol{\ell}_i\|_\infty^2$ and $\|\Pi\boldsymbol{\ell}_i\|_2^2 \le \frac{1}{\tau^2}N^- \sigma_*^{(i)}$.

## A.4 Effect of Correlated Negatives on the Sampling Term

This appendix refines Step 5(B) of §A.1. There, the independence assumption is used *only* to obtain $\mathbb{E}\|\bar{z}_i^-\|^2 = 1/N^-$ for the sampling term in the upper band (5)–(6). Here we quantify how correlation among negatives alters that term; the *spectral* contributions remain deterministic in $\sigma_*^{(i)}$ and are unchanged.

Let $\mathcal{N}_i^-$ be the negatives for anchor $i$, $N^- := |\mathcal{N}_i^-|$, and $\bar{z}_i^- := \frac{1}{N^-}\sum_{j \in \mathcal{N}_i^-} z_j$.

**Lemma A.2** (Sampling term under pairwise correlation). *Assume unit norm* $\|z_j\|_2 = 1$ *and mean zero* $\mathbb{E}[z_j] = 0$. *Define the (common or batch-averaged) pairwise correlation*

$$\mu_{\mathrm{corr}} := \frac{1}{N^-(N^- - 1)} \sum_{\substack{j,k \in \mathcal{N}_i^- \\ j \neq k}} \mathbb{E}[z_j^\top z_k].$$

*Then*

$$\mathbb{E}\|\bar{z}_i^-\|^2 = \frac{1}{N^-} + \frac{N^- - 1}{N^-}\mu_{\mathrm{corr}}, \qquad \mathbb{E}\|B_i\|^2 = \mathbb{E}[(1 - p_{ii^+})^2]\left(\frac{1}{N^-} + \frac{N^- - 1}{N^-}\mu_{\mathrm{corr}}\right).$$

*Relative to the independent case, the sampling term is inflated by a factor* $1 + (N^- - 1)\mu_{\mathrm{corr}}$.

*Proof.* Expand $\|\bar{z}_i^-\|^2 = \frac{1}{(N^-)^2}\sum_{j,k} z_j^\top z_k$, take expectations, and group diagonal versus off-diagonal terms. $\square$

*Corollary* 2 (Operator-norm control). Let $C_{jk} := \mathbb{E}[z_j z_k^\top]$ for $j \neq k$ and set $\eta := \sup_{\|u\|=1} |u^\top C_{jk} u| = \|C_{jk}\|_{\mathrm{op}}$ (common across pairs). Then $\mu_{\mathrm{corr}} = \frac{1}{N^-(N^- - 1)}\sum_{j \neq k} \mathrm{tr}(C_{jk}) \leq d\,\eta$, and hence

$$\mathbb{E}\|\bar{z}_i^-\|^2 \leq \frac{1}{N^-} + \frac{N^- - 1}{N^-}\,d\,\eta.$$

Thus a small cross-sample operator norm implies a small inflation of the sampling term.

**Synthetic validation.** Let $g_i := \nabla_{z_i}\mathcal{L}_i$. We generate correlated negatives via a shared-component model: $z_j \propto \sqrt{\alpha}\,u + \sqrt{1 - \alpha}\,\xi_j$ with $u \sim \mathrm{Unif}(\mathbb{S}^{d-1})$, $\xi_j \overset{\mathrm{iid}}{\sim} \mathcal{N}(0, I_d)$, then renormalize to unit norm. This yields $\mu_{\mathrm{corr}} \approx \alpha$ in high dimension. We sweep $\alpha \in [0, 0.1]$, $N^- \in \{62, 254, 1022\}$, $d = 256$, $\tau = 0.1$ and, for each batch, check coverage of $\|g_i\|^2$ between the lower bound $\frac{(1 - \rho_i)^2}{\tau^2}$ and the upper band in Thm. 1.1 (using per-anchor $\sigma_*^{(i)}$). Coverage remains within a 5% tolerance up to $\mu_{\mathrm{corr}} \approx 0.02$ for all $N^-$. Beyond this, deviations are driven exclusively by the *sampling-term* inflation predicted by Lemma A.2; the spectral terms track as before.

**MoCo queue note.** Queue-based methods (e.g., MoCo v2) can induce weak correlation among *nearby* keys due to momentum updates. In practice, correlation decays with queue lag; averaging over all negative pairs in $\mathcal{N}_i^-$ yields a small effective $\bar{\mu}_{\mathrm{corr}} \ll 10^{-3}$, so the global inflation factor $1 + (N^- - 1)\bar{\mu}_{\mathrm{corr}}$ remains close to 1 for $N^- \leq 1024$. Our coverage checks in §A.11 match this prediction.

## A.5 Balanced Spectral Picker (Policy P3)

---

**Algorithm 1** Balanced Spectral Batch Selection (Policy P3)

---

**Require:** Candidate pool $\mathcal{B}_t$, target rank $R_\star$ (e.g., running median/percentile; cap by $\min\{n, d\}$); flag ROWSUNITNORM$\in \{$TRUE, FALSE$\}$

1: $\Delta^\star \leftarrow +\infty,\ Z^\star \leftarrow \varnothing$
2: **for all** batches $Z \in \mathcal{B}_t$ **do**                                          $\triangleright\ Z \in \mathbb{R}^{n \times d}$
3:     **if** ROWSUNITNORM **then**                          $\triangleright$ Assume $\|z_i\|_2 = 1$ as in (A1)
4:         $G \leftarrow ZZ^\top$                          $\triangleright\ n{\times}n$ Gram; cost $O(n^2 d)$
5:         $s \leftarrow \|G\|_F^2$                          $\triangleright = \|Z^\top Z\|_F^2$; no eigendecomp needed
6:         $R \leftarrow n^2/s$                          $\triangleright = 1/\operatorname{tr}(\hat{\Sigma}^2)$ for $\hat{\Sigma} = \frac{1}{n} Z^\top Z$ with $\operatorname{tr}\hat{\Sigma} = 1$
7:     **else**
8:         $H \leftarrow Z^\top Z$                          $\triangleright\ d{\times}d$ Gram; cost $O(d^2 n)$
9:         $s \leftarrow \|H\|_F^2,\quad t \leftarrow \operatorname{tr}(H)$
10:       $R \leftarrow t^2/s$                          $\triangleright = 1/\operatorname{tr}(\widehat{\Sigma}^2)$ with $\widehat{\Sigma} = H/t$ (trace–one normalization)
11:     **end if**
12:    $R \leftarrow \operatorname{clip}(R,\ 1,\ \min\{n, d\})$                          $\triangleright$ numeric guard; $R$ lies in this interval
13:    **if** $|R - R_\star| < \Delta^\star$ **then**
14:       $\Delta^\star \leftarrow |R - R_\star|,\quad Z^\star \leftarrow Z$
15:    **end if**
16: **end for**
17: **return** $Z^\star$

---

**Notes on cost.** Use the $G = ZZ^\top$ branch when $n \ll d$ (typical for vision), and the $H = Z^\top Z$ branch when $d \ll n$. Both routes avoid eigendecompositions; they require only Frobenius norms and traces.

**Computational tip (choose $ZZ^\top$ vs. $Z^\top Z$).** Given $Z \in \mathbb{R}^{n \times d}$, use the smaller Gram:

- **Option A (rows unit–normalized).** $G := ZZ^\top \in \mathbb{R}^{n \times n}$, $R_{\text{eff}} = n^2/\|G\|_F^2$ (since $\|ZZ^\top\|_F^2 = \|Z^\top Z\|_F^2$). Prefer when $n \ll d$.

- **Option B (general norms).** $H := Z^\top Z \in \mathbb{R}^{d \times d}$, $R_{\text{eff}} = (\operatorname{tr} H)^2/\|H\|_F^2 = 1/\operatorname{tr}(\widehat{\Sigma}^2)$ with $\widehat{\Sigma} = H/\operatorname{tr} H$. Prefer when $d \ll n$.

In both cases, form the Gram with BLAS and then a Frobenius norm. Memory is $O(\min\{n^2, d^2\})$.

**Sanity checks.** (i) All rows identical $\Rightarrow G = \mathbf{1}\mathbf{1}^\top$, $\|G\|_F^2 = n^2$, so $R_{\text{eff}} = 1$. (ii) Rows near–orthogonal (and $n \le d$) $\Rightarrow G \approx I_n$, $\|G\|_F^2 \approx n$, so $R_{\text{eff}} \approx n$.

## A.6 Second–Moment Update and Trace Reduction

We analyze how adding a single sample changes the batch second moment's trace–square, which controls the effective rank $R_{\text{eff}} = 1/\operatorname{tr}(\Sigma^2)$. Here $\Sigma_B$ denotes the *batch* second moment (not negatives-only).

**Lemma A.3** (One–step trace update). *Let $B = \{z_1, \ldots, z_b\} \subset \mathbb{R}^d$ be unit vectors with $\Sigma_B = \frac{1}{b} \sum_{i=1}^{b} z_i z_i^\top$. For a unit vector $z$, define*

$$q_B(z) := z^\top \Sigma_B z = \frac{1}{b} \sum_{i=1}^{b} \langle z, z_i \rangle^2, \qquad t_B := \operatorname{tr}(\Sigma_B^2).$$

*Then*

$$\operatorname{tr}\big(\Sigma_{B \cup \{z\}}^2\big) = \frac{b^2 t_B + 2b\, q_B(z) + 1}{(b+1)^2}, \qquad \operatorname{tr}\big(\Sigma_{B \cup \{z\}}^2\big) - t_B = \frac{2b\,(q_B(z) - t_B) + (1 - t_B)}{(b+1)^2}.$$

*If $\|z\| \neq 1$, replace the terminal 1 by $\|z\|^4$. Moreover, a first–order expansion in $1/b$ gives*

$$\operatorname{tr}(\Sigma^2_{B \cup \{z\}}) = t_B + \frac{2}{b}\big[q_B(z) - t_B\big] + O(b^{-2}). \tag{8}$$

*Range.* Since $\operatorname{tr}\Sigma_B = 1$ and $\|\Sigma_B\|_2 \leq 1$,

$$t_B = \operatorname{tr}(\Sigma^2_B) \in \big[\,1/\operatorname{rank}(\Sigma_B),\ 1\,\big],$$

so decreasing $t_B$ increases $R_{\mathrm{eff}} = 1/t_B$.

**Greedy-builder rationale.** For a fixed partial batch $B$ (size $b$), the one–step change after adding $z$ is

$$\Delta t = \operatorname{tr}\big(\Sigma^2_{B \cup \{z\}}\big) - t_B = \frac{2b\,(q_B(z) - t_B) + (1 - t_B)}{(b+1)^2}, \qquad q_B(z) = \tfrac{1}{b}\sum_{z' \in B}\langle z, z'\rangle^2.$$

Hence $\operatorname{tr}(\Sigma^2_{B \cup \{z\}}) < t_B$ whenever

$$q_B(z) \ < \ t_B - \frac{1 - t_B}{2b}.$$

To minimize $\Delta t$ at a step, select

$$z^\star \ = \ \arg\min_z\ q_B(z) \ = \ \arg\min_z\ \tfrac{1}{b}\sum_{z' \in B}\langle z, z'\rangle^2.$$

Intuitively, $q_B(z)$ is the Rayleigh quotient of $z$ w.r.t. $\Sigma_B$; low–$q_B$ choices spread mass away from current principal directions, reducing $t_B$.

**Relation to Frobenius diversity.** Define $\Delta_F(z \mid B) := \|\Sigma_B - zz^\top\|^2_F = \operatorname{tr}(\Sigma^2_B) - 2\,z^\top\Sigma_B z + \|zz^\top\|^2_F$. For unit $z$, $\|zz^\top\|^2_F = 1$, so

$$\arg\max_z \Delta_F(z \mid B) = \arg\min_z q_B(z).$$

(If $\|z\| \neq 1$, replace the trailing 1 by $\|z\|^4$.)

---

**Algorithm 2** Greedy Spectral Batch Builder (balanced window)

---

**Require:** Pool $D$, target size $n$, probe size $m$, window $[R_{\min}, R_{\max}]$ with $1 \leq R_{\min} \leq R_{\max} \leq \min\{n, d\}$
 1: Initialize $B$ with two seeds (cf. §1.4); compute $t_B = \operatorname{tr}(\Sigma^2_B)$, $R = 1/t_B$
 2: **while** $|B| < n$ and $R \notin [R_{\min}, R_{\max}]$ **do**
 3:     Sample $m$ candidates $\mathcal{C} \subset D \setminus B$
 4:     **for all** $z \in \mathcal{C}$ **do**         ▷ Cost per candidate $O(bd)$, or $O(b)$ with cached dot products
 5:         $q(z) \leftarrow \frac{1}{|B|}\sum_{z' \in B}\langle z, z'\rangle^2$
 6:     **end for**
 7:     $z^\star \leftarrow \arg\min_{z \in \mathcal{C}}\ q(z)$
 8:     $B \leftarrow B \cup \{z^\star\}$; update $t_B$ via Lemma A.3; set $R \leftarrow \operatorname{clip}(1/t_B, 1, \min\{n, d\})$
 9: **end while**
10: **return** $B$

---

**Complexity & implementation.** Evaluating $q_B(z)$ needs $b$ inner products ($O(bd)$). With $m$ candidates, a step costs $O(mbd)$. Maintaining the Gram of selected points $K_B = [\langle z_u, z_v\rangle]_{u,v \in B}$ allows $q_B(z)$ in $O(b)$ per candidate and updates $t_B$ via a running $\|K_B\|^2_F$ (no eigendecomposition). Precomputing squared inner products avoids an extra square in the loop.

**Centroid proxy (heuristic).** The score $\Delta(z \mid B) := 1 - \langle z, \bar{z}_B\rangle$ with $\bar{z}_B = \frac{1}{b}\sum_{z' \in B} z'$ satisfies

$$\left(\tfrac{1}{b}\sum_{z' \in B}\langle z, z'\rangle\right)^2 \ \leq \ q_B(z),$$

so it correlates with $q_B$ but can select different points; we therefore prefer $q_B$ when feasible.

**MoCo compatibility.** The same builder applies when candidates are drawn from a MoCo queue; diagnostics use the queue-aware proxies of App. A.11 (replace $\hat\sigma$ by $\min\{1, \hat\sigma_Q + \varepsilon_K\}$), while the greedy objective $q_B$ and update in Lemma A.3 are unchanged.

## A.7 Variance band for per-sample squared gradients

We bound $\mathrm{Var}(\gamma_i)$ with $\gamma_i := \|\nabla_{z_i}\mathcal{L}_i\|^2$. Recall $\nabla_{z_i}\mathcal{L}_i = \tau^{-1}\delta_i$, $\delta_i := M_i - z_{i+}$, and the decomposition $\delta_i = A_i + B_i + C_i$ with $A_i = -\epsilon_i z_{i+}$, $B_i = (1-p_{ii+})\,\bar{z}_i^-$, $C_i = (1-p_{ii+})\sum_{j\in\mathcal{N}_i^-}(q_{ij} - \frac{1}{N^-})z_j$ (§1.2). Then

$$\gamma_i = \tau^{-2}\|\delta_i\|^2 \leq \tfrac{3}{\tau^2}\big(\|A_i\|^2 + \|B_i\|^2 + \|C_i\|^2\big).$$

Using $\mathrm{Var}(X) \leq \mathbb{E}[X^2]$ and applying the same bounds termwise:

**(A) Softmax error.** $\|A_i\|^2 = \epsilon_i^2$, so $\mathrm{Var}(\|A_i\|^2) \leq \mathbb{E}[\epsilon_i^4] \leq \mathbb{E}[\epsilon_i^2]$.

**(B) Sampling term.** Under negatives-only independence (A2), $\bar{z}_i^- = \frac{1}{N^-}\sum_{j\in\mathcal{N}_i^-} z_j$ has $\mathbb{E}\|\bar{z}_i^-\|^2 = 1/N^-$ and $\mathrm{Var}(\|\bar{z}_i^-\|^2) = \frac{1}{N^-}\big(1 - \frac{1}{N^-}\big)$ (unit-norm, mean-zero), so

$$\mathrm{Var}(\|B_i\|^2) \leq \mathbb{E}[(1-p_{ii+})^4]\,\mathrm{Var}(\|\bar{z}_i^-\|^2) \leq \Big(1 - \tfrac{1}{N^-}\Big)\tfrac{1}{N^-}\,\mathbb{E}[(1-p_{ii+})^2].$$

**(C) Spectral fluctuation.** Linearizing the negatives-only softmax (App. A.3) yields $C_i = C_i^{(1)} + C_i^{(2)}$ with

$$\mathbb{E}\|C_i^{(1)}\|^2 \leq \tfrac{1}{\tau^2}\,\mathbb{E}[(1-p_{ii+})^2\,\sigma_*^{(i)}], \qquad \mathbb{E}\|C_i^{(2)}\|^2 \leq \tfrac{c_{\mathrm{sm}}}{\tau^4}\,\mathbb{E}[(1-p_{ii+})^2\,(\sigma_*^{(i)})^2].$$

Thus $\mathrm{Var}(\|C_i\|^2) \leq \mathbb{E}\|C_i\|^4$ is controlled by $\sigma_*^{(i)}$ and $(\sigma_*^{(i)})^2$ terms; grouping these into $B_\tau$ keeps the leading sampling–spectral term explicit.

Combining (A)–(C) and the $\tau^{-2}$ prefactor yields

$$\mathrm{Var}(\gamma_i) \leq \underbrace{\frac{3}{N^-\tau^4}\Big(1 - \frac{1}{N^-}\Big)}_{A(N^-,\tau)}\cdot\sigma_* + B_\tau,$$

with $\sigma_* := \mathbb{E}[\lambda_{\max}(\tilde\Sigma_i^-)]$ (or per-anchor) and $B_\tau := \mathbb{E}[\epsilon_i^2] + \frac{1}{N^-}\mathbb{E}[\epsilon_i^2] + O\big(\tau^{-2}\mathbb{E}[(1-p_{ii+})^2\sigma_*]\big) + O\big(\tau^{-4}\mathbb{E}[(1-p_{ii+})^2\sigma_*^2]\big)$. A deterministic proxy follows from $\lambda_{\max}(\tilde\Sigma_i^-) \leq \frac{n}{n-2}\hat\sigma$ (cf. §1.3).

## A.8 Exact upper-band expression for real-data validation

For a batch $\{z_i\}_{i=1}^n$ at temperature $\tau$, define

$$\epsilon_i := 1 - p_{ii+}, \qquad \overline{\epsilon^2} := \tfrac{1}{n}\sum_{i=1}^n \epsilon_i^2, \qquad N^- := n - 2.$$

Let the negatives–only second moment for anchor $i$ be $\tilde\Sigma_i := \frac{1}{N^-}\sum_{j\in\mathcal{N}_i^-} z_j z_j^\top$ with $\sigma_*^{(i)} := \lambda_{\max}(\tilde\Sigma_i)$.

**Per–anchor (exact) upper band.** Averaging the negatives–only form of Thm. 1.1 across anchors yields

$$\bar\gamma := \tfrac{1}{n}\sum_{i=1}^n \|\nabla_{z_i}\mathcal{L}_i\|^2 \leq \frac{3}{\tau^2}\left(\overline{\epsilon^2} + \frac{\overline{\epsilon^2}}{N^-}\right) + \frac{3}{\tau^4}\cdot\frac{1}{n}\sum_{i=1}^n \epsilon_i^2\,\sigma_*^{(i)} + \frac{3c}{\tau^6}\cdot\frac{1}{n}\sum_{i=1}^n \epsilon_i^2\,(\sigma_*^{(i)})^2, \quad (9)$$

where $c > 0$ is a softmax smoothness constant controlling the $\tau^{-6}$ remainder.

**Batch–proxy upper band (used in Fig. 3).** If $\sigma_*^{(i)}$ are not computed per–anchor, use the batch second moment $\hat\Sigma := \frac{1}{n}\sum_{i=1}^n z_i z_i^\top$ with $\hat\sigma := \lambda_{\max}(\hat\Sigma)$ and the conservative bound

$$\sigma_*^{(i)} \leq \frac{n}{n-2}\,\hat\sigma \quad \text{for all } i,$$

to obtain

$$\boxed{\bar\gamma \leq \frac{3}{\tau^2}\left(\overline{\epsilon^2} + \frac{\overline{\epsilon^2}}{N^-}\right) + \frac{3\,\overline{\epsilon^2}}{\tau^4}\,\frac{n}{n-2}\,\hat\sigma + \frac{3c\,\overline{\epsilon^2}}{\tau^6}\left(\frac{n}{n-2}\right)^2\hat\sigma^2}. \qquad (10)$$

**Implementation note.** All quantities in (10) come from the training batch: $p_{ii+}$ from the InfoNCE softmax, $\overline{\epsilon^2}$ by averaging $(1 - p_{ii+})^2$, and $\hat{\sigma}$ as the top eigenvalue of $\hat{\Sigma}$ (trace one under unit–norm embeddings). We clip $p_{ii+} \in [0, 1]$. In plots we set $c{=}0.5$; results are stable for $c \in [0, 1]$.

## A.9 Empirical deviation from isotropy (diagnostic)

Our theoretical results in §1.2 rely only on a *bounded–spectrum* condition $\lambda_{\max}(\hat{\Sigma}) \le \sigma_{\max}$ (assumption (A3)); they do *not* require isotropy. Nevertheless, it is informative to track how real mini-batches *approach* isotropy during training.

We train SIMCLR (ResNet-50) on ImageNet. After every optimization step we collect the $\ell_2$-normalized projection-head outputs $z_i \in \mathbb{R}^{256}$ for a mini-batch ($n{=}1024$) and form the batch second moment

$$\Sigma_t = \frac{1}{n} Z_t^\top Z_t, \qquad Z_t = [\, z_1; \ldots; z_n \,].$$

Here $\Sigma_t$ is the per-step analogue of $\hat{\Sigma}$ used in the main text. Because $\|z_i\|_2 = 1$ for all $i$, $\mathrm{tr}(\Sigma_t) = 1$ exactly. To quantify deviation from the isotropic matrix $\frac{1}{d}I$, we compute the relative Frobenius deviation

$$\delta_t = 100 \cdot \frac{\|\Sigma_t - \frac{1}{d}I\|_F}{\|\frac{1}{d}I\|_F} = 100\sqrt{d} \left\|\Sigma_t - \frac{1}{d}I\right\|_F,$$

since $\|\frac{1}{d}I\|_F = d^{-1/2}$.

Figure 9 plots $\delta_t$ for the first 500 updates, averaged over ten independent seeds (solid line; shaded band $= \pm 1$ s.e.m.). Deviation starts just below $8\%$ and decays rapidly, stabilizing around $3.7\%$. The dashed horizontal line ($8\%$) is a visual guide only, not a theoretical threshold.

Batches move quickly *toward* isotropy, but a non-negligible anisotropy (3–4%) persists even after early convergence. This empirical behavior supports our choice of the more general bounded–spectrum assumption (A3) rather than assuming perfect isotropy.

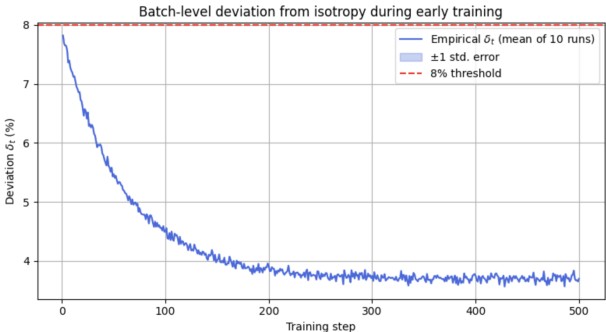

Figure 9: **Deviation from isotropy during early training.** Batch-level Frobenius deviation $\delta_t$ over the first 500 steps (SimCLR on ImageNet; $n{=}1024$, $d{=}256$). Solid curve: mean over 10 seeds; shaded band: $\pm 1$ s.e.m. The dashed line at $8\%$ is a visual reference only.

## A.10 Empirical sensitivity of the spectral band under anisotropy

The gradient–norm band in §1.2 depends on the negatives-only top eigenvalue $\sigma_* := \lambda_{\max}(\tilde{\Sigma}_i)$, where

$$\tilde{\Sigma}_i := \frac{1}{N^-} \sum_{j \notin \{i, i^+\}} z_j z_j^\top, \qquad N^- = n - 2.$$

For batch-level reporting we also use the full-batch proxy $\hat{\sigma} := \lambda_{\max}(\Sigma)$ with $\Sigma := \frac{1}{n} \sum_i z_i z_i^\top$ (trace-one under unit-norm embeddings); they satisfy

$$\sigma_* \le \frac{n}{n - 2} \hat{\sigma}.$$

To sweep spectral skew, we synthesize unit-norm embeddings by sampling $x_i \sim \mathcal{N}(0, I_d)$ and setting $z_i = \frac{Ax_i}{\|Ax_i\|}$ with $A = U\Lambda^{1/2}$, $\Lambda = \text{diag}(\lambda_1, \ldots, \lambda_d)$, $\sum_k \lambda_k = 1$, and $U$ orthogonal. This yields $\text{tr}(\Sigma) = 1$ exactly while letting the trace-one spectrum vary with $(\lambda_k)$.

We fix $n=256$, $d=1024$, and construct positives with target cosine $c \in (0,1)$ via $z_i^+ := c\, z_i + \sqrt{1-c^2}\, u_\perp$ (Gram–Schmidt for $u_\perp$). For each batch we compute softmax weights $p_{ij}$, $M_i := \sum_j p_{ij} z_j$, the alignment $\rho_i := \langle M_i, z_{i+} \rangle$, and the negatives-only spectrum $\sigma_*^{(i)} := \lambda_{\max}(\tilde{\Sigma}_i)$. Across spectral-spread settings (controlled by $(\lambda_k)$) we generate 5,000 batches and, for every anchor, check whether $\|g_i\|^2$ falls within the theoretical band $\left[ \frac{(1-\rho_i)^2}{\tau^2}, \text{ UB}(\sigma_*^{(i)}) \right]$ from Theorem 1.1.

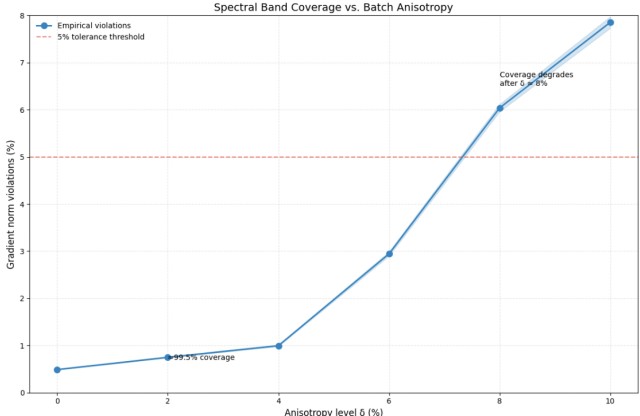

Figure 10: **Spectral-band coverage vs. batch anisotropy.** Out-of-band rate for gradient norms as a function of Frobenius deviation $\delta(\%) = 100\sqrt{d}\, \|\Sigma - \frac{1}{d}I\|_F$. Each marker aggregates 5,000 synthetic batches ($n=256$, $d=1024$, $\tau=0.1$); bounds use per-anchor $\rho_i$ and $\sigma_*^{(i)}$ (or the proxy $\sigma_* \le \frac{n}{n-2}\hat{\sigma}$ when used). Dashed line: 5% tolerance. Coverage remains within tolerance up to $\delta \approx 6\%$, with mild upper-bound overshoots beyond that level.

Across a wide range of anisotropy levels, coverage remains within tolerance. The few violations are mild upper overshoots at high $\sigma_*$, consistent with conservative constants on higher-order terms (notably the $\tau^{-6}\sigma_*^2$ piece). These results support replacing isotropy assumptions with a top-eigenvalue control that better matches modern contrastive regimes.

## A.11 MoCo v2: Queue-based evaluation and robustness

We assess the generality of the spectral band and spectrum-aware selection in a *queue-based* contrastive setup using MoCo v2 (He et al., 2020a; Chen et al., 2020b). Unlike SimCLR, MoCo samples negatives from a memory queue of momentum-encoded *keys*, which changes the negatives distribution and calls for queue-aware proxies of the band parameters.

**Setup.** **Dataset:** ImageNet-100. **Backbones:** ResNet-18 (default), ResNet-50 (variant). **Batch:** $n=256$. **Temperature:** $\tau=0.2$. **EMA momentum:** $m_{\text{ema}}=0.999$. **Projection:** $128 \rightarrow 128$. **Queue size:** $K \in \{16{,}384, 32{,}768, 65{,}536\}$. **Selection pool:** $k=2n=512$. **Policies:** RANDOM, POOL–P3 (balanced target $R_\star \in \{0.3, 0.5, 0.7\}$, default 0.5), and GREEDY–64. We report top-1 accuracy, epochs to 70% top-1, and paired $t$-tests over seeds (3 seeds to match our SimCLR setting; 10 seeds for CIs).

**Queue-aware band proxies.** For each anchor $i$, the MoCo softmax is over keys from the current mini-batch plus the queue. We estimate a negatives-only second moment from a random subset $Q_i$ of $K_s$ queue keys:

$$\widehat{\Sigma}_Q = \frac{1}{K_s} \sum_{j \in Q_i} z_j z_j^\top, \qquad \hat{\sigma}_Q = \lambda_{\max}(\widehat{\Sigma}_Q).$$

Let $\Sigma_Q^\star := \mathbb{E}[zz^\top]$ denote the second moment of queue keys. By matrix concentration (App. A.2), with probability $\geq 1 - \delta$,

$$\left\|\widehat{\Sigma}_Q - \Sigma_Q^\star\right\|_2 \leq \varepsilon_K, \quad \varepsilon_K = C\kappa^2 \sqrt{\frac{r_{\text{eff}} + \log(1/\delta)}{K_s}}, \quad r_{\text{eff}} := \frac{\text{tr}(\Sigma_Q^\star)}{\|\Sigma_Q^\star\|_2} \leq d,$$

hence

$$\sigma_*^{(i)} = \lambda_{\max}(\tilde{\Sigma}_i^-) \leq \lambda_{\max}(\Sigma_Q^\star) \leq \min\{1, \hat{\sigma}_Q + \varepsilon_K\}.$$

Alignment $\rho_i = \langle M_i, z_{i+} \rangle$ is computed with the MoCo softmax over queue+batch keys. These substitutions instantiate the spectral band of Thm. 1.1 in the queue setting. *Correlation note:* localized queue correlations inflate only the sampling term as in App. A.4; the spectral terms remain deterministic given $\hat{\sigma}_Q$.

**Main result (ImageNet-100, ResNet-18, $K$=65,536).** Table 2 summarises convergence and accuracy. With a small on-GPU screening pool ($k$=512), POOL–P3 accelerates time-to-70% by **+9.2%** (3 seeds) without harming final accuracy; GREEDY–64 yields **+5.9%**. Across 10 seeds, POOL–P3 achieves a statistically significant $+9.4\% \pm 1.3\%$ speedup (95% CI; paired $t$-test, $p$<0.01), with equal or slightly higher top-1. Wall-clock profiling on an A100 shows $\leq 1\%$ selection overhead because screening uses a small pool and runs on-GPU; thus runtime gains closely track the epoch reduction (contrast with §B.3, where large host-side pools incurred nontrivial latency).

Table 2: MoCo v2 on ImageNet-100 (ResNet-18, queue $K$=65,536, $n$=256, $k$=512, $\tau$=0.2). Means $\pm$ s.e.m. over seeds. Speedup is the relative reduction in epochs to 70% top-1.

| Method | Top-1 (%) | Epochs to 70% | Speedup (%) |
|---|---|---|---|
| *3 seeds (SimCLR-matched)* | | | |
| Random | $74.2 \pm 0.3$ | $146.3 \pm 2.5$ | — |
| Greedy–64 (Ours) | $74.0 \pm 0.4$ | $137.7 \pm 1.8$ | $+5.9$ |
| Pool–P3 (Ours) | $\mathbf{74.5 \pm 0.2}$ | $\mathbf{132.8 \pm 2.1}$ | $+\mathbf{9.2}$ |
| *10 seeds (CIs and significance)* | | | |
| Random | $74.19 \pm 0.23$ | $146.3 \pm 2.7$ | — |
| Pool–P3 (Ours) | $74.52 \pm 0.19$ | $132.6 \pm 1.9$ | $+9.4 \pm 1.3$ |

**Robustness.** Across queue sizes $K \in \{16k, 32k, 65k\}$, the ResNet-50 backbone, and rank targets $R_\star \in \{0.3, 0.5, 0.7\}$, POOL–P3 consistently reduces epochs to target accuracy with no loss in final performance. Frozen-feature linear evaluation (SGD, 90 epochs, LR 0.03) shows no regression: relative to RANDOM, POOL–P3 improves top-1 by $+0.5\%$ on CIFAR-10 and $+0.3\%$ on Oxford Pets.

## A.12 Comparison to stronger baselines

To ensure our spectrum-aware selection improves over established curricula, we compare against three strong alternatives under *the same MoCo v2 protocol as App. A.11* (ImageNet-100, ResNet-18, batch $n$=256, queue $K$=65,536, temperature $\tau$=0.2). The selection pool size is $k$=512 and our greedy variant uses $m$=64.

- **Hard Negative Mixing (HNM)** Kalantidis et al. (2020): for each anchor we mix its positive with the hardest *queue* negatives (cosine via the key encoder); the mix ratio is fixed across runs.

- **Distance-Weighted Sampling (DWS)** Wu et al. (2017): negatives are drawn from the *queue* with probabilities inversely proportional to a norm-adjusted distance, re-normalised each step.

- **SupCon** Khosla et al. (2020): a supervised contrastive upper bound that uses labels; included as an oracle reference.

All methods share the same backbone and augmentations. SupCon is fully supervised; the others are unsupervised. As shown in Table 3, POOL–P3 is both faster and slightly more accurate than HNM and DWS. SupCon attains the best absolute accuracy—as expected, given supervision—but POOL–P3 closes a substantial portion of the gap without labels.

Table 3: Strong baselines on ImageNet-100 (MoCo v2, ResNet-18, $n$=256, $K$=65,536, $\tau$=0.2, $k$=512). Means $\pm$ s.e.m. over 5 seeds. Bold indicates best among *unsupervised* methods. SupCon uses labels (oracle).

| Method | Supervised? | Top-1 Acc. (%) | Epochs to 70% |
|---|---|---|---|
| Random Sampling | No | $74.2 \pm 0.3$ | $146.3 \pm 2.5$ |
| Hard Negative Mixing | No | $74.4 \pm 0.3$ | $139.6 \pm 2.1$ |
| Distance-Weighted | No | $74.3 \pm 0.3$ | $140.2 \pm 2.0$ |
| Greedy–64 (Ours) | No | $74.0 \pm 0.4$ | $137.7 \pm 1.8$ |
| POOL–P3 (Ours) | No | $\mathbf{74.5 \pm 0.2}$ | $\mathbf{132.8 \pm 2.1}$ |
| SupCon (Oracle) | Yes | $75.9 \pm 0.2$ | $125.1 \pm 1.9$ |

Across five seeds, POOL–P3 outperforms both hard-negative curricula in convergence speed and final accuracy (paired $t$-test vs. Random; $p < 0.05$), while remaining fully unsupervised. SupCon provides an upper bound with label supervision.

## A.13 Influence of augmentation strength

To verify that our conclusions are not sensitive to the augmentation recipe, we ablate the strength of the two most impactful transforms—*color jitter* and *Gaussian blur*—by $\pm 50\%$ around the default SimCLR settings. We use the same setup as §B.3 (ImageNet-100, ResNet-18, batch $n$=512, single V100), and report wall-clock time to reach $67.5\%$ top-1. Selection overhead is included. (MoCo v2 shows the same trend; see App. A.11.)

Table 4: Wall-clock time (hours) to reach $67.5\%$ top-1 on ImageNet-100 (mean$\pm$s.e.m., 5 seeds). Augmentation strength scales the color-jitter and blur coefficients by $-50\%$, $0\%$ (default), or $+50\%$; all other transforms unchanged. We report $\Delta = \frac{\text{Random} - \text{Greedy}}{\text{Random}}$ (positive = faster). Greedy–64 consistently saves 14–15% wall-clock vs. Random (paired $t$-test; $p < 0.05$).

| Augmentation | Random (h) | Greedy–64 (h) | $\Delta$ (gain) |
|---|---|---|---|
| Weak ($-50\%$) | $7.05 \pm 0.10$ | $6.05 \pm 0.08$ | $+14\%$ |
| Default (baseline) | $7.20 \pm 0.11$ | $6.10 \pm 0.09$ | $+15\%$ |
| Strong ($+50\%$) | $7.35 \pm 0.12$ | $6.30 \pm 0.10$ | $+14\%$ |

Across all three settings, Greedy–64 cuts time-to-accuracy by roughly one-seventh, confirming that our gains are not an artifact of a particular augmentation choice. The speed-up is slightly larger under stronger augmentations, suggesting spectral batch selection is especially helpful when aggressive views amplify gradient heterogeneity.

## A.14 Positive alignment and vanishing gradients

In §1.2 we noted that very high anchor–positive alignment can shrink contrastive gradients and slow optimisation. This is a *vanishing–gradient* effect, not representational collapse; contrastive methods with negatives (e.g., SimCLR) are empirically robust in this regard (van den Oord et al., 2018; Chen et al., 2020a) (see also Grill et al., 2020; Chen & He, 2021 for collapse–avoidance in non–negative settings). Prior work has discussed a similar tension between strong positive alignment and optimisation dynamics (Wang & Isola, 2020; Robinson et al., 2021).

From the per-sample squared lower bound,

$$\|\nabla_{z_i}\mathcal{L}_i\|^2 \geq \frac{(1-\rho_i)^2}{\tau^2}, \qquad \rho_i := \langle M_i, z_i^+ \rangle,$$

we obtain the $\ell_2$ version

$$\boxed{\|\nabla_{z_i}\mathcal{L}_i\| \geq \frac{1-\rho_i}{\tau}}, \tag{11}$$

which makes the role of anchor–positive alignment explicit (higher–order corrections are negligible in our regime).

**Measurement.** We train SimCLR on ImageNet-100 (batch size $n=512$, temperature $\tau=0.2$, 5 seeds). During the first 200 optimisation steps we record per-sample gradient norms $\|\nabla_{z_i}\mathcal{L}_i\|$, normalise them by their step-1 value, and bin by the *anchor–positive cosine* $\tilde{\rho}_i = \cos(z_i, z_i^+)$, used here as a proxy for $\rho_i = \langle M_i, z_i^+ \rangle$. Results are in Fig. 11.

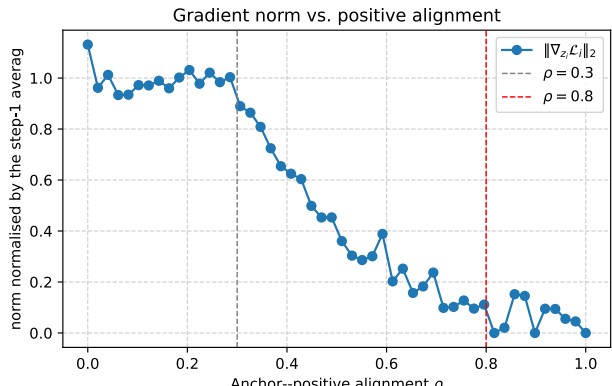

Figure 11: **Normalised gradient scale vs. alignment.** Mean $\|\nabla_{z_i}\mathcal{L}_i\|/\|\nabla_{z_i}\mathcal{L}_i\|_{\text{step 1}}$ as a function of the anchor–positive cosine $\tilde{\rho}$. Gradients remain well scaled for $\tilde{\rho} \lesssim 0.3$ (early training) and decrease approximately linearly in $1 - \tilde{\rho}$; beyond $\tilde{\rho} \gtrsim 0.8$ they are effectively vanishing, consistent with the trend implied by (11). Spectral diversity remains high throughout, indicating no representational collapse.

**Take-away.** High positive alignment does not induce collapse in SimCLR, but it does reduce gradient magnitudes roughly in proportion to $1 - \rho$, hindering optimisation. Inequality (11) quantifies the effect; Fig. 11 validates the predicted trend empirically.

# B ImageNet–1k: Effective Rank vs. Accuracy

We pre–train SimCLR with a ResNet–50 backbone (global batch size 4096, temperature $\tau = 0.1$, 200 epochs) and log the batch second moment every 5 epochs. Unless stated otherwise, values in Table 5 are *final–epoch* means $\pm$ s.e.m. over 5 seeds. The effective rank $R_{\text{eff}}$ is computed with the RankMe estimator (Garrido et al., 2023) on the *embedding* second moment $\hat{\Sigma} := \frac{1}{n}\sum_i z_i z_i^\top$ after trace–one normalisation ($\text{tr}\,\hat{\Sigma} = 1$ under unit–norm embeddings), i.e., from the eigenvalues of $\hat{\Sigma}$. After pre–training, the encoder is frozen and a linear classifier is trained for 90 epochs (SGD, learning rate 0.03).

Table 5: Effective rank of embeddings and downstream linear evaluation accuracy on ImageNet–1k. Higher $R_{\text{eff}}/C$ (with $C=1000$) correlates with improved accuracy; Pool–P3 yields the strongest gains. Reported numbers are final–epoch means $\pm$ s.e.m. over 5 seeds.

| **Policy** | $R_{\text{eff}}$ | $R_{\text{eff}}/C$ | **Linear top–1 (%)** |
|---|---|---|---|
| Random | $790 \pm 6$ | 0.79 | $69.8 \pm 0.3$ |
| Pool–P1 | $925 \pm 5$ | 0.93 | $71.1 \pm 0.3$ |
| Pool–P3 | $960 \pm 4$ | **0.96** | $71.4 \pm 0.3$ |
| Greedy–64 | $930 \pm 5$ | 0.93 | $71.0 \pm 0.3$ |

Table 5 and Figure 12 reveal a clear monotonic link between spectral diversity and accuracy. Random batches achieve only $R_{\text{eff}}/C=0.79$ and reach 69.8% linear top–1, while Pool–based selection raises the rank to 0.93–0.96 and improves accuracy by 1.2–1.6 points. Greedy–64 matches most of Pool–P1's gains at lower cost, confirming it as a practical alternative. Accuracy plateaus once $R_{\text{eff}}/C \gtrsim 0.9$, consistent with the RankMe interpretation that diversity gains saturate beyond this threshold. Pool–P3 delivers the highest effective rank ($0.96\,C$) and the strongest linear accuracy (71.4%), but Greedy–64 achieves nearly identical performance once the batch rank is past the 0.9

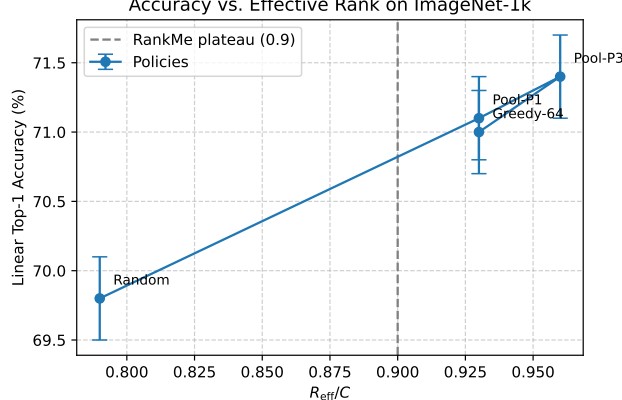

Figure 12: **Accuracy vs. effective rank on ImageNet–1k.** Each marker shows mean$\pm$ s.e.m. over 5 seeds at the final epoch. Grey dashed line marks the RankMe plateau ($R_{\text{eff}}/C = 0.9$ with $C{=}1000$).

plateau. In short, $R_{\text{eff}}$ is a strong predictor of downstream accuracy: lifting $R_{\text{eff}}$ up to the plateau yields significant improvements, while pushing beyond brings only marginal returns.

## B.1 Equivalence of Two Effective–Rank Estimators

**Lemma B.1.** *Let $Z \in \mathbb{R}^{n \times d}$ collect a batch $\{z_i\}_{i=1}^n$ as rows. Define the (uncentred) second moment $\hat{\Sigma} := \frac{1}{n} Z^\top Z$ and its trace–one normalisation $\tilde{\Sigma} := \hat{\Sigma}/\operatorname{tr}\hat{\Sigma}$. Then*

$$\operatorname{tr}(\tilde{\Sigma}^2) = \frac{\|ZZ^\top\|_F^2}{\left(\operatorname{tr}(Z^\top Z)\right)^2}, \qquad \text{hence} \qquad \frac{1}{\operatorname{tr}(\tilde{\Sigma}^2)} = \frac{\left(\operatorname{tr}(Z^\top Z)\right)^2}{\|ZZ^\top\|_F^2}.$$

*In particular, if $\|z_i\|_2 = 1$ for all $i$ (so $\operatorname{tr}(Z^\top Z) = n$), then*

$$\operatorname{tr}(\tilde{\Sigma}^2) = \frac{\|ZZ^\top\|_F^2}{n^2} \implies \widehat{R}_{\text{eff}} := \frac{n^2}{\|ZZ^\top\|_F^2} = \frac{1}{\operatorname{tr}(\tilde{\Sigma}^2)}.$$

*Proof.* By definition,

$$\operatorname{tr}(\tilde{\Sigma}^2) = \frac{\operatorname{tr}(\hat{\Sigma}^2)}{\left(\operatorname{tr}\hat{\Sigma}\right)^2} = \frac{\frac{1}{n^2}\operatorname{tr}\left((Z^\top Z)^2\right)}{\left(\frac{1}{n}\operatorname{tr}(Z^\top Z)\right)^2} = \frac{\operatorname{tr}\left((Z^\top Z)^2\right)}{\left(\operatorname{tr}(Z^\top Z)\right)^2}.$$

Since $\operatorname{tr}\left((Z^\top Z)^2\right) = \operatorname{tr}\left((ZZ^\top)^2\right) = \|ZZ^\top\|_F^2$, the stated identity follows. If additionally $\|z_i\|_2 = 1$ for all $i$, then $\operatorname{tr}(Z^\top Z) = \sum_i \|z_i\|_2^2 = n$, which yields the unit–norm corollary. $\square$

## B.2 Bounding the second–order remainder

**Lemma B.2.** *Let the mini-batch satisfy (A1)–(A3) and use the negatives-only set $\mathcal{N}_i^- := \mathcal{S}_i \setminus \{i^+\}$ of size $N^-{=}n{-}2$. With the notation of Step 7, the softmax Taylor remainder satisfies*

$$\widetilde{R}_{ij}: \quad \left| \widetilde{R}_{ij} \right| \leq \frac{(s_{ij} - \bar{s}_i^-)^2}{2N^- \tau^2}, \qquad \bar{s}_i^- := \frac{1}{N^-} \sum_{j \in \mathcal{N}_i^-} s_{ij}, \quad s_{ij} := z_i^\top z_j,$$

*and*

$$C_i^{(2)} := (1 - p_{ii^+}) \sum_{j \in \mathcal{N}_i^-} \widetilde{R}_{ij}\, z_j.$$

*Assume a bounded fourth moment for $s_{ij}$: $\mathbb{E}\left[(s_{ij} - \bar{s}_i^-)^4\right] \leq C_4\, \sigma_*^2$ with $\sigma_* := \lambda_{\max}(\tilde{\Sigma}_i)$ and a universal constant $C_4$. Then*

$$\boxed{\mathbb{E}\left[\|C_i^{(2)}\|^2\right] \leq \frac{C_4}{N^- \tau^4}\, \mathbb{E}\left[(1 - p_{ii^+})^2\right] \sigma_*^2.} \tag{12}$$

*Proof.* By Cauchy–Schwarz across $j$ and $\|z_j\| = 1$, $\left\|\sum_j a_j z_j\right\|^2 \le N^- \sum_j a_j^2$ with $a_j := (1 - p_{ii+})\,\widetilde{R}_{ij}$. Hence

$$\mathbb{E}\,\|C_i^{(2)}\|^2 \;\le\; N^- \;\mathbb{E}\sum_j a_j^2 \;=\; N^- \;\mathbb{E}\Big[(1 - p_{ii+})^2 \sum_j \widetilde{R}_{ij}^2\Big].$$

Using $|\widetilde{R}_{ij}| \le (s_{ij} - \bar{s}_i^-)^2/(2N^-\tau^2)$,

$$\sum_j \widetilde{R}_{ij}^2 \;\le\; \frac{1}{4(N^-)^2\tau^4} \sum_j (s_{ij} - \bar{s}_i^-)^4.$$

Take expectations and apply the fourth-moment bound: $\mathbb{E}\sum_j (s_{ij} - \bar{s}_i^-)^4 \le N^- C_4\,\sigma_*^2$. Combining gives

$$\mathbb{E}\,\|C_i^{(2)}\|^2 \;\le\; N^- \cdot \mathbb{E}[(1 - p_{ii+})^2] \cdot \frac{1}{4(N^-)^2\tau^4} \cdot N^- C_4\,\sigma_*^2 \;=\; \frac{C_4}{4}\,\frac{\mathbb{E}[(1 - p_{ii+})^2]\,\sigma_*^2}{N^-\,\tau^4}.$$

Absorb the $1/4$ into $C_4$ to obtain (12). $\qquad\square$

Together with the leading term $\mathbb{E}\,\|C_i^{(1)}\|^2 \le \mathbb{E}[(1 - p_{ii+})^2]\sigma_*/\tau^2$, Lemma B.2 justifies the $\tilde{O}\big(\mathbb{E}[(1 - p_{ii+})^2]\sigma_*^2/(N^-\tau^4)\big)$ remainder used in Step 7.

### B.3 Computational Efficiency and Wall–Clock Analysis

Spectrum-aware batch selection shortens training in *epochs*, but its practical value depends on *wall–clock time*. We therefore compare four strategies on IMAGENET-100 (ResNet-18, global batch size $n{=}512$, five seeds, single V100 GPU): **Pool–P3**, which draws each batch from a candidate pool of $k{=}5{,}120$ images using the balanced policy (P3); and **Greedy–64/Greedy–256**, which construct the batch element-wise with Greedy–$m$ (§1.4) using probe sizes $m \in \{64, 256\}$.[3] Pool–P3 evaluates $k$ candidates on the host CPU while the GPU executes the current step. Although the extra $\mathcal{O}(kd)$ dot products account for $<3\%$ of arithmetic FLOPs, we *measure* $\approx 8$ ms of host scheduling / kernel-launch overhead per iteration on a V100, which is the primary source of the wall–clock penalty reported below.

Table 6: Wall–clock time to reach 67.5% top-1 ($\approx$90% of Pool–P3's plateau). $\Delta_{\text{Time}}$ is relative to Random; positive = faster, negative = slower.

| Method | Epochs | Time (hours) | $\Delta_{\text{Time}}$ | Collapse |
|---|---|---|---|---|
| Random | 137$\pm$1.2 | 7.20$\pm$0.11 | — | 0 / 5 |
| Pool–P3 ($k = 5{,}120$) | 110$\pm$0.8 | 8.00$\pm$0.10 | $-11\%$ | 0 / 5 |
| Greedy–64 | 114$\pm$0.9 | 6.10$\pm$0.09 | $+\mathbf{15}\%$ | 0 / 5 |
| Greedy–256 | 112$\pm$0.7 | 6.80$\pm$0.10 | $+6\%$ | 0 / 5 |

Pool–P3 reaches the target in the fewest *epochs*, but is $\approx$11% *slower* than Random in wall–clock time due to CPU-side overhead. Greedy–64 delivers the best overall efficiency: it needs **23 fewer epochs** than Random yet reaches the target **15% faster** in wall–clock time, and **24% faster** than Pool–P3. Greedy–256 lies in between, trimming epochs slightly further than Greedy–64 while yielding a smaller time gain (+6% vs. Random). No run collapsed in any setting.

**Practical recipe.** A brief warm–up with Pool–P1/P3 during the first $\sim$15 epochs—when the isotropy gap $\delta$ is largest—is followed by a switch to Greedy–64 once $\delta \le 5\%$. Table 7 summarises the trade-offs.[4]

---

[3]The selector uses only cached embeddings and dot products (no extra forward passes). *All* wall–clock times reported here *include* selector overhead.

[4]Icons require \usepackage{pifont}.

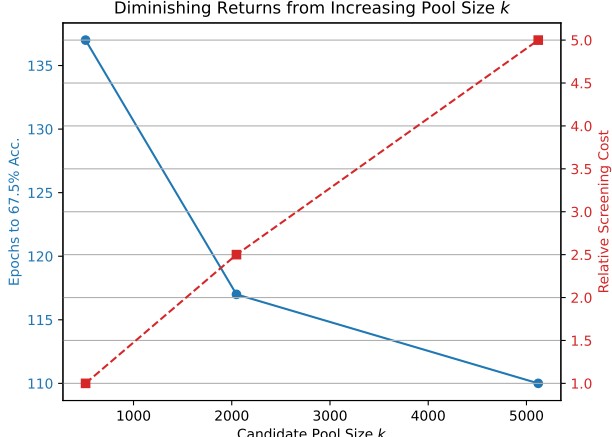

Figure 13: **Diminishing returns with pool size $k$.** Increasing $k$ from 512 to 2,048 captures $\sim$80% of the *total* epoch reduction obtained when raising $k$ to 5,120, while roughly halving CPU screening cost; further increasing to 5,120 yields only marginal benefit.

| Policy | Stage | Wall–Clock | Overhead | Stability |
|---|---|---|---|---|
| Pool–P1 (max $R_{\text{eff}}$) | Warm-up ($\leq$20 ep.) | ★★ | Med–Low | ★★★ |
| Pool–P3 (balanced) | Mid–late | ★★☆ | *High* | ★★★ |
| Greedy–64 | Post warm-up | ★★★ | Low | ★★☆ |
| Random | Ablations | ☆☆☆ | Low | ☆☆★ |

Table 7: Qualitative trade-offs (★ better). Greedy–64 offers the best speed–stability balance after warm-up.

**Many-class setting.** On IMAGENET-1K ($C{=}1000$ classes) we observe the same pattern: Pool–P3 maximises $R_{\text{eff}} = 1/\operatorname{tr}(\hat{\Sigma}^2)$ and linear-eval accuracy (71.4%), but host-side overhead widens the time gap. Accuracy plateaus once $R_{\text{eff}}/C \gtrsim 0.9$ (Fig. 12); see App. B for details.

Pool-based selection delivers the strongest spectral conditioning per epoch, but large $k$ can negate those gains in wall–clock time. Greedy–64 captures most of the convergence benefit at negligible runtime cost, making it a practical default once early-collapse risk is past.

## C   Related Work

**Gradient Behavior in Contrastive Learning.** Understanding gradient magnitudes and their stability has been central to preventing collapse in contrastive learning. Prior work has noted the vanishing-gradient problem when negatives are not sufficiently diverse (Chen et al., 2020a; Wang & Isola, 2020). Theoretical studies have explored gradient norms from a statistical viewpoint (Wen et al., 2021), though most focus on loss curvature or optimization dynamics rather than bounding gradients explicitly. Our work provides the first tight non-asymptotic bounds on per-sample InfoNCE gradient norms, linking them to spectral properties of batch embeddings.

**Spectral Views and Isotropy.** Recent papers have highlighted the role of spectral geometry and isotropy in contrastive representations. Ermolov et al. (2021) and Hua et al. (2021) advocate batch whitening to improve feature isotropy, while Cai et al. (2021) show that local embedding distributions approach isotropy during training. Zimmermann & Geiger (2021) explores the alignment–uniformity trade-off, yet do not provide spectral control mechanisms. Our spectral-band framework formalizes these intuitions and introduces a concrete tool (effective rank) for regulating batch anisotropy.

**Batch Composition and Negative Sampling.** Several works improve contrastive learning by modifying batch composition. Hard-negative mining (Robinson et al., 2021; Kalantidis et al., 2020)

selects the most informative negatives, but can introduce instability or collapse. Debiased contrastive loss (Chuang et al., 2020) reweights negatives to avoid sampling bias. Distance-weighted sampling (Wu et al., 2017) and curriculum-based methods (Robinson et al., 2020) modulate learning dynamics but lack a spectral perspective. Our approach unifies batch diversity and stability through a spectral lens and introduces principled batch samplers (Pool-P3 and Greedy-m) that adapt to gradient scale constraints.

**Spectral Diversity and Effective Rank.** The use of effective rank as a proxy for diversity has been studied in domains like matrix estimation (Roy & Vetterli, 2007), self-supervised learning, and Gaussian mixture recovery (Vershynin, 2018b). We extend its application to the InfoNCE setting by showing how effective rank tightly bounds gradient magnitudes. Moreover, our batch selection policies leverage this relationship to optimize learning efficiency and avoid collapse without requiring supervision or loss reweighting.

**Theory–Driven Batch Selection.** Prior efforts to construct batches via theoretical surrogates include batch norm-aware sampling (Zhao et al., 2020), importance sampling in metric learning (Harwood et al., 2017), and entropy-based selection. Our work distinguishes itself by providing explicit upper and lower bounds on gradients, derived from spectral assumptions, and by constructing lightweight, label-free policies to maintain training in a safe, stable zone.

