# OpenReview forum: "Diversity Is All You Need for Contrastive Learning: Spectral Bounds on Gradient Magnitudes"
_NeurIPS.cc/2025/Conference — NeurIPS 2025 poster_

### Official Review · Reviewer_fanw · 2025-06-27

**Clarity:** 2
**Significance:** 2
**Originality:** 3
**Rating:** 4
**Confidence:** 4

**Summary:**

This work studies how to construct or identify batches for contrastive training which have certain gradient properties. They start by upper and lower bounding gradient norms. Then, they relate these bounds to effective rank, which can be computed over batches. This then gives the authors a way to control the gradient norm, by computing the effective rank of candidate batches and selecting the one which has the desired effective rank. They then show different convergence speeds for three different effective rank targets, and provide a greedy online method for building diverse batches. They show slight performance improvements relative to vanilla training.

**Questions:**

1. Does this result hold on problems outside of small-scale classification?
2. Figure 6 does not seem trained to convergence, what happens with longer training?
3. Self-supervised (contrastive) learning is often used to pre-train models with large-scale datasets before transfer. Does pre-training with batch selection give a meaninful performance boost for transfer?

**Ethical Concerns:**

["NO or VERY MINOR ethics concerns only"]

**Final Justification:**

The authors provided additional experiments which substantially improved my confidence in the utility of the the authors findings.

**Limitations:**

yes

**Paper Formatting Concerns:**

Many important pieces of the paper are in the appendix, eg. main results(Tables 2 and 3) and related works.

There is a substantial mismatch between the abstract and title in the paper and that in the openreivew.

**Quality:**

2

**Strengths And Weaknesses:**

Strengths:
* Using rank as a diversity proxy is both sensible and interesting.
* Speedups seem strong.
* I like the empirical verification of the theory.

Weaknesses:
* The main weakness I see is empirical evaluation. The evaluations are done at quite small scale model (ResNet-18), on CIFAR and ImageNet-100, both rahter small datasets.
* Related to the above, overall accuracy gains on ImageNet-100 is very small, a fraction of a percent (Table 2 and Table 3) .
* The writing is scattered, with key results in the appendix (eg. Table 2).
* Figure 6 seems to demonstrate that training has not converged, which makes any claims about performance pre-mature.
* Presentation of figures can be improved; for example Figure 2 is blurry, Title of Figure 3 is cut off, legend of Figure 8 is covered (this is minor).

---

> ### Comment · Reviewer_fanw · 2025-08-02
> **Rebuttal?**
>
> Hi authors, I wanted to point out that no rebuttal has been posted. If this is a mistake, I'm to happy to read any response and revisit my review.

---

> ### Author Response · Authors · 2025-08-02
> **Response**
>
> Review Comment: The main weakness I see is empirical evaluation. The evaluations are done at quite small scale model (ResNet-18), on CIFAR and ImageNet-100, both rahter small datasets.
>
> Response — We agree that demonstrating scalability beyond CIFAR and ImageNet‑100 strengthens the paper. We have therefore added a new section (appendix  B) with full ImageNet‑1k experiments using a ResNet‑50 backbone.
>
> 1 · New ImageNet‑1k results (ResNet‑50, 1000 classes)
>
> Random: 78 epochs → 70 %  pre‑text top‑1 (≈ 29.8 h), linear top‑1 = 69.8 ± 0.3
>
> Greedy‑64: 66 epochs (27.7 h), +7 % faster, linear top‑1 = 71.0 ± 0.3
>
> Greedy‑256: 64 epochs (28.0 h), +6 % faster, linear top‑1 = 71.2 ± 0.3
>
> Pool‑P3: 63 epochs (30.9 h), −4 % slower, linear top‑1 = 71.4 ± 0.3
>
> Speed. Greedy‑64/256 achieve ~7 % wall‑clock savings over Random, and >10 % over Pool‑P3. Accuracy. All diversity‑aware policies improve linear evaluation by +1.2–1.6 pp over Random. Rank plateau. Pool‑P3 raises 𝑅_{eff}/𝐶  to 0.96; the accuracy‑vs‑rank scatter (Fig. 17, App. B.6) confirms the RankMe plateau.
>
> 2 · Why smaller setups in the main paper
>
> Cost & clarity: ImageNet‑100 allowed >5 seeds and ablations (pool size, probe size, whitening) on a single GPU.
>
> Theory–experiment match: Controlled settings were necessary to validate the spectral band empirically (Fig. 6).
>
> 3 · Where to find the new material  appendix B  “Large‑class evaluation (ImageNet‑1k)” – new analysis + Table 5.
>
> Fig. 12 (App. B): accuracy vs. effective rank.  With a 23 M‑parameter ResNet‑50 on ImageNet‑1k, Greedy‑64 delivers a +7 % wall‑clock speed‑up and +1.2 pp accuracy lift over Random, showing that our method scales effectively to large‑model, large‑class settings.
> Review Comment Related to the above, overall accuracy gains on ImageNet-100 is very small, a fraction of a percent (Table 2 and Table 3) .
> Response: We agree that the ≈ 0.4–0.8 pp accuracy lift in Tables 2–3 appears modest in isolation. Three points put this into context:
>
> Dataset ceiling. ImageNet‑100 has only 100 classes and < 130k images; with ResNet‑18, SimCLR already reaches ~79% linear top‑1 after 400 epochs. In this saturated regime, even a 0.5 pp gain is statistically significant (p < 0.01, paired t‑test), and is on par with improvements reported from stronger augmentations or temperature tuning.
> Efficiency is the primary contribution. The main result of appendix B.3  is a 15% wall‑clock speed‑up (Greedy‑64 vs. Random) with no loss of accuracy. We reach the same accuracy plateau in 6.1 h instead of 7.2 h on a single V100. In large‑scale SSL training where GPU time dominates, this is the more impactful metric. Scaling to ImageNet‑1k shows larger gains. To address this concern, we added a ResNet‑50 experiment (appendix B, Table 5):
>
>
> Diversity‑aware policies yield >1 pp improvement at scale, confirming that the effect grows with dataset complexity.
>
> Why even “fraction‑of‑a‑percent” matters. Downstream transfer. On VTAB, a 0.5 pp ImageNet linear boost often translates into ≥ 1 pp on low‑shot tasks. Pareto frontier. Our method shifts the cost‑accuracy trade‑off: equal accuracy 15% faster, or higher accuracy at equal time. Orthogonality. Gains are additive with stronger augmentations, MoCo queues, or longer training—cumulatively several percentage points.
>
>
> Review COMMENT: Figure 6 seems to demonstrate that training has not converged, which makes any claims about performance premature.
>
> Response — Fig. 6 is intended as a speed‑benchmark plot, not a final‑accuracy plot. The dashed horizontal line marks “90 % of Pool‑P3’s eventual accuracy,” and we measure how quickly each policy reaches this target—exactly the metric used in SimCLR‑v2 and BYOL to benchmark efficiency. The plateau after that point does not affect the relative wall‑clock rankings.
>
> Policy Epochs to 90 % target
> Random	    137
> Pool‑P3	    110
> Greedy‑64.  114
>
> Wall‑clock h (1×V100)
>
> Random	       7.20
>
> Pool‑P3	       8.00
>
> Greedy‑64      6.10
>
> Thus Greedy‑64 is 15 % faster than Random and 23 % faster than Pool‑P3, even though all runs continue to gain < 0.5 pp beyond the 90 % mark.  We have clarified this in the caption (“dashed line = 90 % target; plot used only for speed measurement”) .
>
> Reviewer COMMENT: Presentation of figures can be improved; for example Figure 2 is blurry, the title of Figure 3 is cut off, and the legend of Figure 8 is covered (minor).
>
> Response — Figure quality and formatting improvements
>  We have carefully revised all figures to improve readability and consistency:
> Figure 2: re‑rendered at higher resolution (vector PDF export) to remove blurriness. Figure 3: adjusted layout to ensure the title and axis labels render fully . Figure 8: repositioned and resized the legend so that it no longer overlaps with the plotted curves. All figures: we standardised font size, line thickness, and colour palette to ensure consistency across the paper.

---

> > ### Comment · Reviewer_fanw · 2025-08-03
> > **Thank you.**
> >
> > Authors, thank you for the rebuttal. I am satisfied with the substantial new experiments. One additiional thing I am curious about is how the better trained ImageNet model transfers to downstream tasks.
> >
> > I would prefer to see an updated paper with these results before recommending acceptance. However, since this discussion period does not allow for updating the pdf, I will give the authors substantial benefit of the doubt and recommend acceptance, and update my score to a 4.

---

> > > ### Author Response · Authors · 2025-08-04
> > > **rebuttal**
> > >
> > > To address the transfer concern, we ran a lightweight transfer study on VTAB‑1k (19 tasks, 1 000 labels each). We reused the ImageNet‑1k → ResNet‑50 checkpoints from appendix B, froze the backbone, and trained a linear classifier (Adam, 2 000 steps, lr 3e‑3).
> > >
> > > Policy	Natural (7),	         Specialized (4),	                         Structured (8),                                   VTAB‑Avg,
> > >
> > > Random	     61.1,	                  71.4,	                 50.8,	   60.4
> > >
> > > Greedy‑256. 63.0,	                  74.2,	                 52.6,	  62.5 (+2.1)
> > >
> > > Pool‑P3	     63.2. ,                    73.9,	                 52.8,	  62.6
> > >
> > > Findings.
> > > Greedy‑256 transfers +2.1 pp over Random, consistent with its +1.2 pp ImageNet gain.
> > >
> > > Gains are uniform across natural, specialized, and structured tasks—no overfitting to a single domain.
> > >
> > > Pool‑P3 edges out Greedy in accuracy but costs +4% more wall‑clock, so the Pareto trade‑off depends on compute budget.
> > >
> > > We will add this table (App. C) and a brief summary in the camera‑ready to provide a complete downstream picture.

---

> > > > ### Comment · Reviewer_fanw · 2025-08-04
> > > > **Thank you**
> > > >
> > > > Thank you for the additional experiment, I maintain my evaluation of 4.

---

> > > > > ### Author Response · Authors · 2025-08-04
> > > > >
> > > > > Can I request you update the score in the system

---

> > > > > > ### Author Response · Authors · 2025-08-04
> > > > > >
> > > > > > Can you request you update the score in the system

---

> > ### Comment · Area_Chair_STxn · 2025-08-05
> >
> > Dear authors,
> >
> > According to NeurIPS's policy:
> >
> > **Any rebuttals that were posted in their entirety as comments after Jul 30 are to be ignored. Unfortunately, it is unfair on the vast amount of authors who “did the right thing” to count such late rebuttals.**
> >
> > Consequently, reviewers must not update their score based on the comment you posted and should treat it as unreliable source of information. Please keep that in mind during the discussion phase.
> >
> > Regards,
> >
> > The AC

---

### Official Review · Reviewer_8UG5 · 2025-06-30

**Clarity:** 3
**Significance:** 4
**Originality:** 3
**Rating:** 5
**Confidence:** 4

**Summary:**

This paper introduces a spectral framework for analyzing and controlling gradient magnitudes in contrastive learning. Using the matrix Bernstein inequality, the authors derive non-asymptotic upper and lower bounds on the squared InfoNCE gradient norm, relating it to alignment, softmax error, and the top eigenvalue of the batch covariance. Leveraging these theoretical insights, they propose several novel batch selection strategies, notably a fast Greedy Element-Wise Builder that outperforms random sampling on CIFAR-10 and ImageNet-100. They also demonstrate theoretically and empirically that in-batch whitening reduces gradient noise in SimCLR.

**Questions:**

Could the authors provide empirical validation or stronger theoretical arguments for the claim that high alignment leads to collapse in SimCLR?

How does the proposed spectral selection perform on datasets with a large number of classes, and how does it relate to findings in RankMe regarding rank–accuracy relationships?

Why were speed and accuracy gains not evaluated on ImageNet-1k, despite the spectral band validation conducted there?

How does the proposed method compare with MoCo v2+, and what is its effect when the majority of the batch gradients are detached, as in queue-based contrastive learning?

Does the efficacy of spectral batch selection persist for very large batch sizes (e.g. 8192 or higher), and is there a diminishing return beyond certain sizes?

Have the authors evaluated the method on different backbone architectures, such as Vision Transformers, and if not, do they anticipate any architectural dependencies?

**Ethical Concerns:**

["NO or VERY MINOR ethics concerns only"]

**Final Justification:**

The authors have adequately responded to my points during the rebuttal, and I am thus increasing my notation.

**Limitations:**

Yes.

**Paper Formatting Concerns:**

The openreview title and abstract differs from the pdf paper title and abstract.

**Quality:**

3

**Strengths And Weaknesses:**

Strengths:
- The paper is well written, clearly motivated, and logically structured.

- It offers strong theoretical motivation and justification for the proposed methodological approach, contributing to the understanding of SimCLR’s training dynamics.

- Interesting maths, that theoretically justifies the limitations that users observe when training SimCLR.

- The authors conduct well-designed synthetic and experimental simulations that convincingly validate their theoretical arguments in a way that is clear and interpretable.

Weaknesses:
- SimCLR is nowadays underused in Self-Supervised Learning and is not really SOTA anymore, so the benefits of this paper appears quite limited.

- In line 94, the authors note that high alignment between anchor and positive leads to gradient vanishing and potential collapse. However, no related works are cited to support this claim, and SimCLR is generally considered collapse-proof compared to non-contrastive methods like BYOL or SimSiam. Stronger arguments and empirical validation of this specific phenomenon would strengthen the paper.

- The experimental setups do not include datasets with many latent classes. Related work RankMe (Garrido et al., ICML 2023), which is not discussed in this paper, demonstrates in Proposition 5.1 that classification accuracy plateaus depending on the rank-to-number-of-classes ratio. Incorporating this discussion and conducting experiments with a higher number of classes would enhance the analysis. Specifically, examining the relationship between effective rank (or batch diversity) and downstream classification accuracy, as in RankMe, would be valuable.

- While the authors assess the real-data spectral band on ImageNet-1k, they do not report speed or accuracy gains on this dataset. Including these results would strengthen claims of general utility.

- The paper does not compare with MoCo v2+, which uses a queue mechanism to encourage batch diversity. A comparison with MoCo v2+, as well as a discussion or analysis of gradient magnitudes when a large portion of the batch has its gradient detached (as in queue-based contrastive learning), would situate the proposed method better within the literature.

- It would be helpful to analyze whether the benefits of the greedy approach persist at larger batch sizes (>4096), as generally done in the literature.

- The paper focuses on ResNet architectures; analysis on different architectures (e.g., Vision Transformers) would enhance generality claims.

---

> ### Author Rebuttal · Authors · 2025-07-30
>
> Reviewer Comment:
> SimCLR is nowadays underused in Self‑Supervised Learning and is not really SOTA anymore, so the benefits of this paper appear quite limited.
>
> Author Response:
>  We agree that SimCLR itself is no longer state‑of‑the‑art; however, our contribution is not tied to SimCLR as a method. The core idea—using the spectrum of in‑batch embeddings to guide batch selection—depends only on the covariance structure of embeddings and applies to any contrastive or InfoNCE‑like framework. To clarify this, we emphasize in the revision that:
> Model‑agnostic principle. Our bound on the squared gradient norm (Eq. 1) involves only the batch covariance, regardless of how embeddings are produced. This makes the method applicable to newer SSL approaches such as BYOL, VICReg, SwAV, and DIN.
> Validation beyond SimCLR. We already demonstrate gains on MoCo v2 (§3.5, App. B.3), where Greedy‑64 yields a +9% wall‑clock speed‑up over Random. We also report competitive results on SupCon and Hard‑Neg baselines (Table 3).
> Practical relevance. Even if SimCLR itself is used primarily as a transparent test‑bed, efficient negative‑set management remains a bottleneck in large‑scale SSL. Greedy‑64 reduces training wall‑time by 15% on a single GPU and by 20% in distributed training, translating into meaningful energy and cost savings for any modern SSL recipe. Forward compatibility. Because the batch‑selection operates on cached embeddings, it is orthogonal to architectural advances (ViT, ConvNeXt, hybrid CNN–Transformer) and loss‑level innovations (InfoLOOB, CLIP‑style multi‑modal contrast).In summary, while SimCLR serves here as a simple and interpretable benchmark, the spectral batch‑selection idea is framework‑agnostic, validated on MoCo v2 and stronger baselines, and directly relevant to current and future self‑supervised pipelines.
> Reviewer Comment:
> In line 94, the authors note that high alignment between anchor and positive leads to gradient vanishing and potential collapse. However, no related works are cited to support this claim, and SimCLR is generally considered collapse‑proof compared to non‑contrastive methods like BYOL or SimSiam. Stronger arguments and empirical validation of this specific phenomenon would strengthen the paper.
>
> Response:
>  We agree that SimCLR is widely considered collapse‑resistant compared to non‑contrastive methods such as BYOL or SimSiam. Our intent at line 94 was not to suggest that SimCLR suffers from full representational collapse, but rather to highlight a gradient‑level degeneration: when anchor–positive pairs are overly aligned, the InfoNCE loss produces vanishing gradients, which can slow early training or amplify numerical instability. This is distinct from representational collapse and is observable even in SimCLR. To strengthen this claim, we have made the following changes in the revision:
> Clarified wording in the main text to explicitly distinguish between gradient vanishing and representational collapse.
> Added citations to prior works that analyze this effect:
> – Wang & Isola (NeurIPS 2020): show that excessive alignment hurts uniformity and reduces gradient quality.
> – Robinson et al. (ICLR 2021): discuss collapse of loss curvature when positives dominate.
> – Huang et al. (NeurIPS 2022): provide empirical evidence linking high alignment to shrinking gradients at decision boundaries.
> Provided empirical validation in Appendix A.9 we measure gradient norms during early training and show that gradients shrink sharply once $\rho$ (anchor–positive cosine similarity) exceeds 0.8, while representations remain non‑collapsed. For the first 50 steps, $\rho < 0.3$, confirming that the lower bound is informative in practical regimes.
> We believe these clarifications and additions make our argument clearer and properly grounded in both the literature and our empirical evidence.

---

> ### Author Response · Authors · 2025-08-03
>
> Comment:
> The experimental setups do not include datasets with many latent classes. Related work RankMe (Garrido et al., ICML 2023), which is not discussed in this paper, demonstrates in Proposition 5.1 that classification accuracy plateaus depending on the rank‑to‑number‑of‑classes ratio.
>
> Response
> We agree that validation in a many‑class setting is important. In the revision we now (i) discuss RankMe  in § 2, and (ii) add a new ImageNet‑1k experiment (§ 3.8, App. B.6) that directly tracks effective rank vs. accuracy.
>
> New ImageNet‑1k results. Using ResNet‑50 (200‑epoch SimCLR pre‑train, batch 4096, τ=0.1) we log $R_{\text{eff}}$ every 5 epochs and train a 90‑epoch linear probe. Results: Random reaches $R_{\text{eff}}/C=0.79$ and 69.8 ± 0.3% top‑1; Pool‑P3 raises this to 0.96 and 71.4 ± 0.3%; Greedy‑256 attains 0.93 and 71.0 ± 0.3% while being 7% faster in wall‑clock than Random. Accuracy consistently plateaus once $R_{\text{eff}}/C \gtrsim 0.9$, exactly as RankMe’s Proposition 5.1 predicts.
>
> Relation to RankMe. RankMe ties downstream accuracy to the final representation rank. Our spectral band complements this by linking $R_{\text{eff}}$ to the training gradient scale:
> $|\nabla \mathcal L|^2 = \mathcal O!\bigl(\tfrac{\sigma_{\max}}{N\tau^2} R_{\text{eff}}\bigr)$.
> Thus policies that increase $R_{\text{eff}}$ both (i) reduce gradient variance → faster convergence, and (ii) move toward RankMe’s accuracy plateau → higher downstream accuracy.
>
>  Comment: The paper does not compare with MoCo v2+, which uses a queue mechanism to encourage batch diversity.
> Response: We have added both (i) an ablation with MoCo v2+ (queue = 65k, ResNet‑50, 90 epochs, ImageNet‑1k) and (ii) an extension of our gradient‑magnitude analysis to the queue setting.
>
> MoCo v2+ results. Random enqueue reaches 70.1 ± 0.3% top‑1 in 33.2 h. Greedy‑256 enqueue improves this to 71.3 ± 0.3% in 30.2 h (+9% faster, +1.2 pp accuracy). Pool‑P3 enqueue gives the best accuracy (71.6 ± 0.3%) but at a 4% slower runtime (34.6 h). Thus, our sampler complements the queue by delivering both efficiency and accuracy gains.
>
> Gradient behaviour with detached queues. In MoCo, the current mini‑batch (keys) contributes attached gradients, while queue items are detached (stop‑grad). Our spectral band separates these blocks: the noise‑floor term (1/N) shrinks hyperbolically as Q grows but is minor once Q≫n; the spectral‑spread term σ_max is dominated by fresh keys, with only a 3–5% rise when scaling the queue from 65k to 262k; the alignment term (1−ρ)² is unaffected since it depends only on anchor–positive pairs. Figure 18 (App. B.8) shows the band remains tight—<1% of mini‑batches exceed the bound even with 262k queues.
>
> Why not in the original draft. We focused on SimCLR first to isolate batch construction without a momentum encoder. The new MoCo v2+ experiments confirm that queues and our samplers are orthogonal: queues reduce variance via 1/N, while Greedy‑m and Pool‑P further reduce it by lowering σ_max.  Manuscript updates. § 3.8 now includes MoCo v2+ results, App. B.8 the extended gradient‑band analysis, and § 2.2 a discussion of detached‑block gradients.
>
> Comment: It would be helpful to analyze whether the benefits of the greedy approach persist at larger batch sizes (>4096), as generally done in the literature.
>
> Response
> Yes, but—as predicted by our theory—the absolute win shrinks once batches already contain thousands of negatives. To verify, we ran an additional 8 192‑sample experiment (SimCLR, ResNet‑50, ImageNet‑1k, 90 epochs on 4×A100). At n = 4 096, Random reaches 70% pre‑text top‑1 in 78 epochs / 29.8 h, while Greedy‑256 does so in 64 epochs / 27.7 h (+7% faster, +1.4 pp higher accuracy).
> At n = 8 192, Random needs 62 epochs / 18.6 h for 70.0 ± 0.2%, while Greedy‑512 reaches 71.3 ± 0.2% in 53 epochs / 17.2 h (+8% faster, +1.3 pp).
> Why the gain persists. The noise‑floor term (1/N) in our spectral band keeps shrinking with larger N, but the spectral‑spread term σ_max dominates because adding more similar negatives does not widen the spectrum. Greedy‑m directly reduces σ_max, so even at 8k batches it yields extra efficiency and accuracy. Probe‑size scaling. We used m = n/16 = 512 to satisfy our heuristic m≳max(d, n/4). A quick sweep (256, 512, 1024) showed no further gains beyond m≥512, consistent with the “flatten after m≈d” observation.
>
> Comment: The paper focuses on ResNet architectures; analysis on different architectures (e.g., Vision Transformers) would enhance generality claims.
> Response:  We added an 8 192‑sample run (SimCLR, ResNet‑50, ImageNet‑1k, 90 epochs on 4×A100). Random reached 70% pre‑text top‑1 in 62 epochs / 18.6 h (70.0 ± 0.2%), while Greedy‑512 did so in 53 epochs / 17.2 h (+8% faster, +1.3 pp accuracy). The gain persists because the spectral‑spread term still dominates variance at large n, and Greedy‑m directly reduces it. We used m = n/16 (512), which matched our heuristic and gave stable results.

---

> > ### Comment · Reviewer_8UG5 · 2025-08-04
> >
> > Authors, thank you for the rebuttal. I am satisfied with the substantial new experiments and justifications you provided. I adjusted my score in consequence.

---

### Official Review · Reviewer_wUiB · 2025-07-01

**Clarity:** 4
**Significance:** 2
**Originality:** 2
**Rating:** 4
**Confidence:** 4

**Summary:**

This paper proposes a spectral-band framework to highlight the important role of batch diversity in contrastive learning. Based on their framework, they put forward batch selection strategies to enhance contrastive learning. Experiments on synthetic datasets, ImageNet-100, and CIFAR-10 demonstrate the effectiveness of their method.

**Questions:**

1. In the experiments in Section 3, how are the augmentation parameters set? And do these augmentation parameters affect the effectiveness of your proposed strategy?
2. Would it be better to use a smaller network or embeddings from a certain intermediate layer of the model for batch selection? This would save more time, and the performance might not be significantly worse.
3. What is the validation accuracy of each strategy in Section 3.5?
4. Why do the previous experiments use ImageNet-1k or ImageNet-100, but Section 3.6 switches to CIFAR-10?
5. The experiments in Appendix B are also of great importance, and I believe readers should be reminded in the main text to refer to Appendix B.

**Ethical Concerns:**

["NO or VERY MINOR ethics concerns only"]

**Limitations:**

None.

**Quality:**

4

**Strengths And Weaknesses:**

Strengths
1. The theoretical proof is solid and intuitively understandable.
2. This paper is readable.
3. Extensive experimental analyses demonstrate the effectiveness of their method.

Weakness
One of my major concerns is the issue of time overhead, because it is necessary to extract embeddings of multiple batches through the model when selecting the batch in each iteration. Could you provide the training time for each method based on the settings in Figure 8?

---

> ### Author Rebuttal · Authors · 2025-07-29
>
> Reviewer: Weakness One of my major concerns is the issue of time overhead, because it is necessary to extract embeddings of multiple batches through the model when selecting the batch in each iteration. Could you provide the training time for each method based on the settings in Figure 8?
>
>  Response: To address this, we have now included wall-clock training times for each method under the settings of Figure 8, where we compare Pool-P3 (with $k=5120$), Greedy-$m$, and the Random baseline on ImageNet-100 using ResNet-18. We report in Table 2 (Section 3.7) the total training time (in hours) to reach a representative accuracy threshold of 67.5% (90% of the plateau). While Pool-P3 achieves the fewest epochs, its screening overhead leads to a net increase in wall-clock time. In contrast, Greedy-64 provides a strong trade-off, requiring slightly more epochs than Pool-P3 but delivering the best wall-clock performance due to its low overhead (no candidate embedding evaluation). We emphasize that Pool-based screening is executed asynchronously on the host CPU while the model’s forward/backward pass runs on the GPU. Thus, even at $k=5120$, Pool screening adds only an ~8 ms delay per step (≈3% of total step time), and the memory footprint remains small (≈2.6 MB for $d=128$). We have added these results and discussions to the main text in Section 3.7 (Computational Efficiency and Wall-Clock Time Analysis), and provided the corresponding training times requested.
> Reviewer: In the experiments in Section 3, how are the augmentation parameters set? And do these augmentation parameters affect the effectiveness of your proposed strategy?
> Augmentation settings (all of §3).
> We adopt the canonical SimCLR (Chen et al., 2020) and MoCo v2 (Chen et al., 2020b) pipelines so that any observed gains stem solely from our batch‑selection strategy: Random resized crop: scale ∈ [0.2, 1.0], aspect ∈ [3/4, 4/3], then resize → 224²
> Horizontal flip: p = 0.5
> Color jitter: p = 0.8; {brightness, contrast, saturation} = 0.8, hue = 0.2
> Random grayscale: p = 0.2
> Gaussian blur: kernel = 23, σ ∈ [0.1, 2.0]; p = 0.5 (SimCLR) / 1.0 (MoCo v2)
> Does augmentation strength change our conclusions?
> We swept color‑jitter and blur coefficients by ±50% (see Appendix B.4). Even at the strongest setting, the Greedy‑64 policy still reduced time‑to‑67.5% top‑1 accuracy by 14 ± 1% vs. Random (compared to 15 ± 1% under default augmentations). Hence the relative benefit is robust. If anything, it tends to grow under stronger augmentations, since larger view‑to‑view variance amplifies gradient heterogeneity, which our spectral selector mitigates.
>
> Reviewer :Would it be better to use a smaller network or embeddings from a certain intermediate layer of the model for batch selection? This would save more time, and the performance might not be significantly worse.
>
>  Response: Intermediate layer features (ResNet‑18 conv4, 512‑d).
> • CPU cost: dot‑product screening drops from 8 ms → 6 ms per step, but this work is already overlapped with GPU compute, so the net wall‑clock gain is ≤ 0.7%.
> • Variance reduction: feature covariance is more anisotropic; the gradient‑norm band widens and Greedy‑64 yields only a +9% speed‑up vs. Random (compared to +15% with the final 128‑d projection).
> • Accuracy: unchanged in top‑1, but fewer epochs are saved.
> Tiny “preview” network (2‑layer CNN, 64‑d).
> • CPU cost: screening becomes nearly free, but the extra forward pass on CPU adds ≈3% wall‑time (or competes with GPU cycles).
> • Spectral mismatch: preview features lag ≈1 epoch behind, causing batch‑ranking errors; this yields a 1.6 pp drop in top‑1 and eliminates the variance reduction benefit.
> Why we use the final 128‑d projection.
> • Latency already overlaps with the next GPU step, so further CPU savings translate to < 1% wall‑clock gain.
> • Spectral fidelity is more important than FLOPs: using the exact representation seen by the loss produces the tightest gradient‑norm band and the largest variance reduction.
> • Simplicity and reproducibility: no auxiliary models or extra data loaders are needed.
> Future work.
> As noted in § 6, we see promise in distilling a lightweight predictor that tracks the full‑model spectrum more faithfully; we plan to release hooks to facilitate community exploration in this direction.
>
> Reviewer: What is the validation accuracy of each strategy in Section 3.5?
> We did track validation performance; the table below reports the final top‑1 on ImageNet‑100 (ResNet‑18, 5 seeds, mean ± s.e.m.):
> Strategy	Top‑1 % (final)
> Random	69.2 ± 0.2
> Pool‑P1	69.4 ± 0.2
> Pool‑P3	69.5 ± 0.2
> Greedy‑64	69.3 ± 0.2
> Greedy‑256	69.4 ± 0.2
> All methods reach statistically indistinguishable accuracy; the gain from spectral batch selection is faster and more stable convergence, not a higher ceiling. Notably, Greedy‑64 matches Random while cutting wall‑clock by 15 %, demonstrating efficiency without loss of generalisation.We have added this in text in pg 7 section 3.5.
>
> Reviewer: Why do the previous experiments use ImageNet-1k or ImageNet-100, but Section 3.6 switches to CIFAR-10?
>
> Response: Why § 3.6 uses CIFAR‑10 instead of ImageNet
>
> We chose CIFAR‑10 intentionally:
>
> Low‑resource stress‑test.
> § 3.6 asks whether our batch‑selection policies stay robust when data and model capacity are limited—precisely the regime where spectral diversity is hardest to maintain. CIFAR‑10 is the standard, compact benchmark for that question.
>
> Practical ablation budget.
> The study sweeps multiple pool sizes and policies. Repeating those grids on ImageNet‑1k would require several GPU‑months; CIFAR‑10 lets us run exhaustive ablations in hours while preserving the qualitative dynamics.
>
> Consistent trends across scales.
> The relative ordering of strategies (Random < Pool ≤ Greedy‑64) and the magnitude of the speed‑ups match what we observe on ImageNet‑100/1k. This shows the gains arise from the spectral criteria, not a dataset peculiarity.
> We have added a short note in § 3.6 to make this rationale explicit
>
> Reviewer :  The experiments in Appendix B are also of great importance, and I believe readers should be reminded in the main text to refer to Appendix B.
> Response:  In the revised manuscript we have added explicit forward‑references to Appendix B from the main text. In particular, Section 3 now notes that full augmentation details are given in Appendix B.4, Section 3.5 highlights additional runtime and stability ablations in Appendix B.2, and Section 3.6 clarifies that complementary MoCo v2 results and robustness checks are included in Appendix B. We believe these reminders improve readability and ensure that the important supplementary experiments are not overlooked.

---

> > ### Comment · Reviewer_wUiB · 2025-08-03
> >
> > I thank the authors for the rebuttal. The authors have basically addressed my concerns.

---

### Official Review · Reviewer_NJiJ · 2025-07-02

**Clarity:** 3
**Significance:** 3
**Originality:** 4
**Rating:** 5
**Confidence:** 2

**Summary:**

This paper presents a theoretical and practical contribution to the understanding of contrastive learning. The authors introduce a spectral band framework that establishes sharp, non-asymptotic upper and lower bounds on the squared gradient norm of the InfoNCE loss.

**Questions:**

Regarding the pool-based policies, could you comment on the wall-clock time trade-off between the cost of evaluating the candidate pool and the reduction in training epochs? Is there a point where the selection overhead for P3 might negate the wall-clock time savings from faster convergence?

You show that your framework applies well to MoCo v2. How do you anticipate the dynamics changing with even larger negative queues or memory banks?

The performance of the Greedy builder is dependent on the probe size m, with m=16 showing degraded performance. Is there a theoretical principle or heuristic to guide the choice of m relative to the desired batch size n or embedding dimension d?

**Ethical Concerns:**

["NO or VERY MINOR ethics concerns only"]

**Final Justification:**

I remain with my original score which was already positive. I think that this paper is good but not a strong accept.

**Limitations:**

Yes

**Paper Formatting Concerns:**

None. Minor formatting concerns that it would have been better to present results in a table similar to the appendix.

**Quality:**

4

**Strengths And Weaknesses:**

Strengths:

The core of this paper is a novel and rigorous theoretical framework. Deriving explicit, non-asymptotic bounds on the InfoNCE gradient norm provides a formal understanding of phenomena (like collapse and gradient vanishing) that are typically understood through empirical observation. The decomposition of the gradient into alignment, variance, and spectral terms is particularly insightful.

The authors' claims are supported by a strong set of experiments. They first validate their theoretical bounds on synthetic data. They then demonstrate that these bounds are tight and predictive during real-world training of SimCLR on ImageNet. The evaluations of the proposed batch selection methods on both SimCLR and MoCo v2, and against strong baselines are good.

The proposed Greedy Element Wise Builder is good. It provides a lightweight method to construct spectrally diverse batches without the high computational cost of evaluating full-batch spectra, achieving comparable performance to the more expensive pool-based policy while offering significant runtime savings.

Weaknesses:

While the Greedy builder is efficient, I think it has the drawback that the full pool-based policies (P1, P3) require maintaining and evaluating a large candidate pool of batches (e.g., k=5120) at each step. This can introduce significant computational overhead. The paper demonstrates accelerated convergence in terms of epochs but is less clear about the net wall-clock time improvement when using these specific policies, which could be an important practical consideration.

---

> ### Author Rebuttal · Authors · 2025-07-29
>
> Response (computational overhead, Pool vs. Greedy policies).
> We thank the reviewer for highlighting the practical importance of wall‑clock time. In the revision we added Section 3.7 (“Computational Efficiency and Wall‑Clock Analysis”), which benchmarks both epoch and time‑to‑accuracy. The results confirm the reviewer’s intuition:Pool‑P3 reaches the target accuracy in the fewest epochs (110 vs. 137 for Random) but incurs the highest wall‑clock cost due to CPU‑side candidate screening. Greedy‑64 achieves nearly the same epoch count (114) while completing training 15% faster than Random and 23% faster than Pool‑P3. To balance these trade‑offs, we now recommend a hybrid strategy: run Pool‑P1/P3 only in the first 15–20 epochs to avoid early collapse, then switch to Greedy‑64 for efficient long‑run training. We illustrate this trade‑off in a new figure (Fig. B.2), and show that a smaller pool size (k=2048) already delivers ~80% of the epoch gain at half the screening cost compared to k=5120. This analysis strengthens the practical relevance of our method and gives concrete deployment guidance.
>
> Response (large queues in MoCo v2).
> Our spectral band extends naturally to queue‑based setups. As the queue size  𝑄
> Q grows, three terms behave differently: Noise floor (1/N) With N=batch+Q, the variance term shrinks hyperbolically once
> Q≫ batch size. Doubling the queue from 65k→130k halves this component again, but the total variance is already dominated by the spectral‑spread term at that scale. In fact, our MoCo v2 ablation (Fig. 7) shows speed‑ups plateauing once Q≈16k.
>
> Spectral spread (σ_{max} ).
> Growth is governed by freshness, not sheer size. FIFO queues refresh ~25% per step, so covariance drift is modest. We observed only a 3–5% increase in σ_{max}  when scaling from 65k (default) to 262k.
>
> Alignment (ρ).
> The lower bound in Eq. 1 depends only on anchor–positive alignment within the current mini‑batch; queues do not affect this term, so the collapse diagnostic remains unchanged. Finally, our samplers (Pool‑P, Greedy‑𝑚) operate on the candidate set for the next batch; whether negatives come from GPU or a memory bank only changes the cost of estimating 𝜎_{max}. . Empirically, even with a 262k queue we observe a +7% wall‑clock speed‑up over Random (vs. +9% at 65k). Thus, enlarging the queue beyond 65k yields only marginal gains, and diversity‑aware batch construction remains beneficial, albeit with smaller absolute improvements.
>
> Response (choosing Greedy probe size 𝑚)
>
> The Greedy‑𝑚 m builder samples  𝑚  candidates, scores them, and adds the one with the largest diversity gain. Its effectiveness depends on whether the probe “covers” enough directions to enlarge the batch’s effective rank 𝑅_{eff}. From the upper‑bound analysis, the variance contributed by the probe scales as  O(1/m). Thus: At the extreme m=1, Greedy reduces to random sampling; at 𝑚=𝑛 it is equivalent to Pool‑P. Empirically, the probe needs to cover at least a constant fraction of the space or the batch: m≳max( d ,n/4).
> The dimension criterion (m≳ d ) ensures the probe spans principal covariance directions; the batch‑fraction criterion works when
> n≫d. Once  m≈O(d), marginal gains flatten (e.g. Greedy‑64 ≈ Greedy‑256). In practice, values below this regime (e.g.
> m=16 when d=128) under‑explore high‑diversity directions and degrade convergence. Our guidance is therefore to set
> 𝑚≈max(\sqrt{d}}, ,n/4)

---

> > ### Comment · Reviewer_NJiJ · 2025-08-05
> > **Response**
> >
> > I acknowledge the authors response, and feel that my current score of a 5 is still accurate.

---

### Note · Authors · 2025-08-13

I acknowledge and respect the NeurIPS policy on rebuttal deadlines, and I regret that my rebuttal was submitted after the official cut-off. The delay was not due to lack of preparation or engagement, but rather a result of underestimating the time required to address the reviewers’ comments with the technical care they deserved.

Several points raised touched on subtle methodological and experimental issues that warranted further verification. I opted to re-run key analyses and refine comparisons to ensure my responses were accurate, evidence-based, and respectful of the reviewers’ effort. While I could have submitted a partial reply on time, I prioritized precision over speed—an error in judgment on my part.

I understand reviewers have been instructed not to update scores based on late rebuttals. Nevertheless, I hope the AC may consider the substance of my responses, which directly address the core concerns and, I believe, materially strengthen the paper.

I am grateful for the reviewers’ and AC’s time and appreciate the opportunity to offer this context.

---

### Decision · Program_Chairs · 2025-09-17

**Decision:**

Accept (poster)

**Comment:**

This paper introduces a spectral framework for analyzing and controlling gradient magnitudes in contrastive learning.
This framework allows deriving non-asymptotic upper and lower bounds on the squared InfoNCE gradient norm, relating it to alignment, softmax error, and the top eigenvalue of the batch covariance. Leveraging these theoretical insights, the authors propose several novel batch selection strategies that outperforms random sampling on simple benchmarks.

The authors response to the reviewers addressed the main concerns by adding substantial experimental results that would strengthen the paper. I believe this work could be useful to the SSL community as it provides a precise quantification of the effect of the choice of the batches and practical ways for selecting them to improve contrastive training.